



# A Twenty-Year Analysis of Winds in California for Offshore Wind Energy Production Using WRF v4.1.2

Alex Rybchuk[1,2], Mike Optis[1], Julie K. Lundquist[1,3], Michael Rossol[1], and Walt Musial[1]

[1]National Renewable Energy Laboratory, Golden, Colorado, USA
[2]Department of Mechanical Engineering, University of Colorado Boulder, Boulder, Colorado, USA
[3]Department of Atmospheric and Oceanic Sciences, University of Colorado Boulder, Boulder, Colorado, USA

**Correspondence:** A. Rybchuk (alex.rybchuk@nrel.gov)

**Abstract.** Offshore wind resource characterization in the United States relies heavily on simulated winds from numerical weather prediction (NWP) models, given the lack of hub-height observations offshore. One such NWP data set used extensively by U.S. stakeholders is the Wind Integration National Dataset (WIND) Toolkit, a 7-year time-series data set produced in 2013 by the National Renewable Energy Laboratory. In this study, we present an update to that data set for offshore California that

leverages recent advancements in NWP modeling capabilities and extends the period of record to a full 20 years. The data set predicts a significantly larger wind resource (0.25–1.75 m s$^{-1}$ stronger), including in three Call Areas that the Bureau of Ocean Energy Management is considering for commercial activity. We conduct a set of yearlong simulations to study factors that contribute to this increase in the modeled wind resource. The largest impact arises from a change in the planetary boundary layer parameterization from the Yonsei University scheme to the Mellor-Yamada-Nakanishi-Niino scheme and their diverging

wind profiles under stable stratification. Additionally, we conduct a refined wind resource assessment at the three Call Areas, characterizing distributions of wind speed, shear, veer, stability, frequency of wind droughts, and power production. We find that, depending on the attribute, the new data set can show substantial disagreement with the WIND Toolkit, thereby driving important changes in predicted power.

*Copyright statement.* This work was authored in part by the National Renewable Energy Laboratory, operated by Alliance for Sustainable

Energy, LLC, for the U.S. Department of Energy (DOE) under Contract No. DE-AC36-08GO28308. Funding provided by the U.S. Department of Energy Office of Energy Efficiency and Renewable Energy Wind Energy Technologies Office. The views expressed in the article do not necessarily represent the views of the DOE or the U.S. Government. The U.S. Government retains and the publisher, by accepting the article for publication, acknowledges that the U.S. Government retains a nonexclusive, paid-up, irrevocable, worldwide license to publish or reproduce the published form of this work, or allow others to do so, for U.S. Government purposes.

## 1 Introduction

California's renewable energy portfolio has steadily grown during the past decade, and in recent years interest has turned toward developing an offshore wind industry (Dvorak et al., 2010; Jacobson et al., 2014; Speer et al., 2016). In depths up to





1,000 m, the outer continental shelf (OCS) off the coast of California has an abundant wind energy resource—approximately 110,000 MW of net technical resource capacity (Musial et al., 2016). Approximately 95% of this resource is found in deeper
OCS waters (60–1000 m) where floating wind turbine technology is required. To date, most offshore wind farms across the globe have been built in shallow waters where fixed-bottom turbines are suitable (shallower than 60 m). Recent technological developments have started to open the door to floating wind turbines that can be deployed in deeper waters (Musial et al., 2019b; Beiter et al., 2020). U.S. state and federal agencies have begun to explore capturing this resource (Gilman et al., 2016). In the OCS, the Bureau of Ocean Energy Management (BOEM) is currently exploring leases for commercial wind energy
development in three Call Areas: Humboldt, Morro Bay, and Diablo Canyon (Fig. 1).

One major challenge with offshore wind energy is uncertainty in the wind resource at wind turbine hub height (Archer et al., 2014; Shaw et al., 2019b). Reduced uncertainty leads to improved design requirements, safer maintenance, and reduced financing costs (Clifton et al., 2016). This uncertainty stems in part from a sparse offshore observational network (Hahmann et al., 2015; Banta et al., 2018), particularly hub-height observations. Until recently, the OCS has lacked public offshore wind
measurements. In October 2020, two floating lidars were deployed off the coast (one near Humboldt, the other near Morro Bay), which measure winds up to approximately 250 m and can be used to characterize winds across a turbine's rotor-swept area. Additionally, observations from the National Oceanic and Atmospheric Administration National Data Buoy Center (NDBC) provide high-quality wind measurements at the surface, but these surface-based observations introduce an additional degree of uncertainty relative to hub-height observations, given how different surface winds can differ from winds aloft.

Because of sparse hub-height observations, offshore wind resource assessments rely heavily on modeled wind speeds from numerical weather prediction (NWP) models, such as the Weather Research and Forecasting (WRF) model (Skamarock et al., 2019). These models provide estimates of hub-height winds at desired sites at high spatial and temporal resolutions (Storm and Basu, 2010; Murthy and Rahi, 2017). Many U.S. offshore wind resource assessments (e.g. Musial et al., 2019a, 2020) have employed the Wind Integration National Dataset (WIND) Toolkit (WTK in figures and tables) (Draxl et al., 2015), developed by
the National Renewable Energy Laboratory (NREL). This data set is a wind resource assessment conducted for the contiguous United States (CONUS) based on 7 years (2007–2013) of WRF simulations. Wind data from the lowest 200 m is stored at 5-minute, 2-km resolution, and it is publicly available through Amazon Web Services as well as NREL's Wind Prospector web-based interface at https://maps.nrel.gov/wind-prospector/.

Since the release of the WIND Toolkit in 2015, NWP models have experienced significant development. For example,
measurements from two major observational campaigns, the Wind Forecast Improvement Project (WFIP, Wilczak et al., 2015) and the Second Wind Forecast Improvement Project (WFIP2, Shaw et al., 2019a), have been used to improve parameterizations within WRF (Olson et al., 2019b) for the specific purpose of improving wind resource modeling and forecasting. Synoptic-scale atmospheric forcing used to drive simulations has been improved with updated reanalysis products, such as the 5th generation European Centre for Medium-Range Weather Forecasts reanalysis product (ERA5, Hersbach et al., 2020) and the second
Modern-Era Retrospective analysis for Research and Applications product (MERRA2, Gelaro et al., 2017). These updated NWP components have been applied in new wind resource assessments, including the New European Wind Atlas (Hahmann et al., 2020; Dörenkämper et al., 2020).



Given advancements in NWP, NREL and BOEM have collaborated to produce an updated 20-year data set, called CA20, that leverages state-of-the-art NWP modeling capabilities in offshore California. Optis et al. (2020) details the development
of this data set and addresses the questions: How does modeled wind resource change as a result of these advancements in NWP, and why do these changes arise? Specifically, the report discusses a series of pre-production simulations that drove the modeling decisions behind CA20, the development of CA20, key differences in modeled winds between CA20 and the WIND Toolkit across the OCS, the sensitivity of CA20 winds with respect to different modeling decisions, and uncertainty in the wind resource.

In this study, we highlight key findings of the technical report, explore additional reasons for variability between CA20 and the WIND Toolkit across the OCS, and analyze differences between the data sets in the BOEM Call Areas. In Sect. 2, we describe the model data sets. In Sect. 3, we discuss the large hub-height wind speed differences between CA20 and the WIND Toolkit, extend the sensitivity study of Optis et al. (2020), and conduct a new temporal analysis regarding winds across the OCS. In Sect. 4, we provide a refined wind resource analysis at the centroids of the three Call Areas, examining distributions
of wind speed, atmospheric stability, wind shear, directional veer, wind droughts, and power output. In Sect. 5, we conclude the report. Although this study focuses primarily on modeled winds, we provide a novel comparison between modeled and observed surface winds at 17 NDBC buoys in Appendix A.

## 2  Methods

The WIND Toolkit was created to study wind energy integration in the power grid, but it has become a popular tool for
wind resource assessment across much of the contiguous United States. The CA20 data set builds upon and shares many characteristics with the WIND Toolkit (Table 1). The WIND Toolkit was developed using a 7-year (2007–2013) simulation with WRF 3.4.1. CA20 builds upon this by using WRF 4.1.2 across a 20-year period (2000–2019). To simulate the full period, both assessments simulated individual months as distinct runs that included two prior days for spin-up. The results from individual months were concatenated to form the final product.

CA20 has two domains that are horizontally nested with one-way communication to the inner domain. The outer domain has a horizontal resolution of 6 km and uses a fixed 30-second timestep. The inner domain, which covers the entire California coastline (Fig. 1), has a horizontal resolution of 2 km and a fixed 10-second timestep. The heights of the 61 vertical levels are specified using a hyperbolic function that provides higher resolution at the surface, with a total of 9 levels between 0–200 m. The WIND Toolkit used 41 vertical levels, 5 of which were within 0–200 m. The Mellor–Yamada–Nakanishi–Niino (MYNN)
planetary boundary layer (PBL) scheme (Nakanishi and Niino, 2009, MYNN) and the MM5-Jiménez surface layer parameterization (Jiménez et al., 2012) are used in the lower atmosphere. These choices are in contrast to the WIND Toolkit, which uses the Yonsei University (YSU) PBL scheme (Hong et al., 2006) and the traditional MM5 surface layer parameterization. The ERA5 reanalysis product, used in CA20, provides atmospheric forcing at an hourly resolution and a 31-km horizontal resolution. ERA-Interim (Dee et al., 2011), used in the WIND Toolkit, forces at 79-km and 6-hour resolution. For sea surface temper-
atures (SSTs), CA20 uses the daily Operational Sea Surface Temperature and Sea Ice Analysis (OSTIA, Donlon et al., 2012)



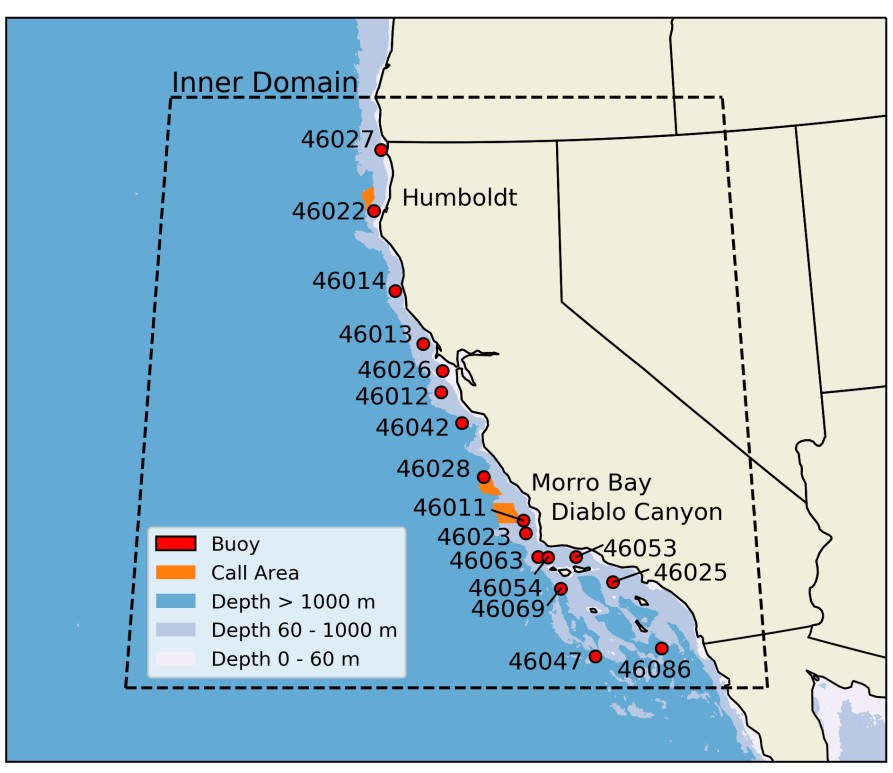

**Figure 1.** Simulation domain with buoys (red) and all Call Areas (orange).

|                                | CA20                     | WIND Toolkit            |
| ------------------------------ | ------------------------ | ----------------------- |
| Domain                         | California               | CONUS                   |
| WRF version                    | 4.1.2                    | 3.4.1                   |
| Period                         | 2000–2019                | 2007–2013               |
| Horizontal resolution          | 6 km, 2 km               | 6 km, 2 km              |
| Vertical levels                | 61 (9 between 0–200 m)   | 41 (5 between 0–200 m)  |
| PBL parameterization           | MYNN                     | YSU                     |
| Surface layer parameterization | MM5-Jiménez              | MM5                     |
| Reanalysis product             | ERA5                     | ERA-Interim             |
| Sea surface temperature        | OSTIA                    | NCEP RTG                |

**Table 1.** Simulation configurations for CA20 and the WIND Toolkit.

data set, which is part of ERA5 and reports SSTs at 1/20 degree resolution, whereas the WIND Toolkit used the daily National
Centers for Environmental Prediction (NCEP) Real-Time Global (RTG) SST 1/12-degree resolution product (Grumbine, 2020).
CA20 applies spectral nudging on a 6-km domain every 6 hours. Additional simulation details include the Kain-Fritsch cumu-





| Sensitivity Simulations |
| --- |
| PBL scheme (YSU) |
| WRF version (3.4.1) |
| Reanalysis product (ERA-Interim) |
| SST product (NCEP RTG) |
| Time period (2007–2013) |
| Domain size (CONUS) |

**Table 2.** List of additional characteristics explored as part of sensitivity analysis.

lus parameterization in the outer domain, eta microphysics, RRTMG for radiation, and the unified Noah land surface model.
The full CA20 namelist with all the simulation parameters can be found at https://doi.org/10.5281/zenodo.45975482280.

Instantaneous CA20 meteorological fields are saved every 5 minutes, and data output speeds are increased by using parallel-netCDF (Li et al., 2003). WRF netCDF output is converted to the HDF5 file format (Folk et al., 2011) to provide quick access to data for analysis and to minimize storage costs. During this process, some quantities (wind speed, wind direction, temperature) are interpolated to heights 10, 40, 60, 80, 100, 120, 140, 160, 180, and 200 m. The surface Obukhov length,
friction velocity, and roughness length are additionally stored. Data analysis was conducted primarily with packages from the Pangeo environment (Odaka et al., 2020), such as Dask (Rocklin, 2015) and Cartopy (Met Office, 2010).

In addition to the 20-year CA20 data set, we conduct several year-long simulations for 2017. In this study, these simulations are used to study the impact of individual factors on updated winds in CA20 (Table 2), but they have also been used to quantify uncertainty in the modeled winds (Optis et al., 2020). These sensitivity simulations are run for the same domain with nearly
identical conditions to CA20 but with only one or two factors changed for comparison to WIND Toolkit conditions. The details regarding each simulation are discussed in Sect. 3.4.

## 3   Hub-Height Mean Winds in CA20 and WIND Toolkit

### 3.1   Overall Mean Winds and Wind Speed Differences

We calculate the *overall* mean hub height wind speeds across the entire CA20 and WIND Toolkit data sets (Fig. 2), recognizing
that CA20 averages from 2000–2019, whereas the WIND Toolkit averages from 2007–2013. CA20 shows substantially stronger hub-height mean winds than the WIND Toolkit nearly everywhere across the OCS. Central California shows the largest gain in overall mean wind speeds, approximately 1.75-m s$^{-1}$. The Diablo Canyon and Morro Bay Call Areas in this region see approximately 1.5 m s$^{-1}$ increases. Northern California also sees substantial increases in wind speeds. Overall mean CA20 winds are approximately 12 m s$^{-1}$ here, and thus the Humboldt Call Area sees an approximately 1-m s$^{-1}$ increase from the
WIND Toolkit.





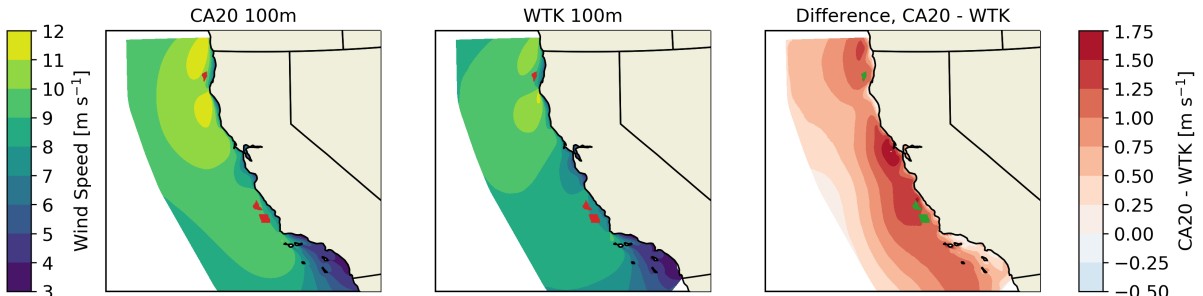

**Figure 2.** Overall 100-m mean winds in CA20 and the WIND Toolkit as well as their differences. Call Areas highlighted in red (left, center) and green (right).

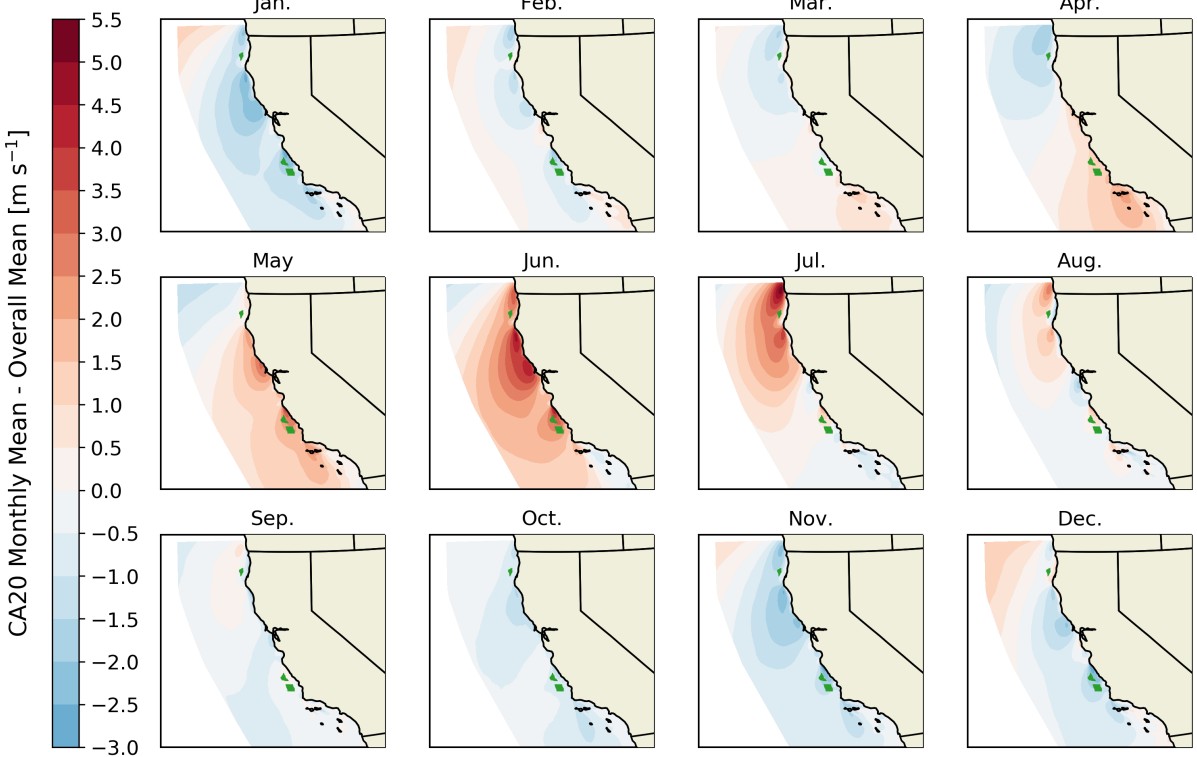

**Figure 3.** Monthly CA20 mean 100-m winds, with reference to the overall CA20 mean.

## 3.2 Annual, Monthly, and Hourly Mean Winds

Average CA20 and WIND Toolkit wind speeds are also calculated on annual, monthly, and hourly scales. To conserve space, key figures are shown here. Additional figures can be found in Appendix B.

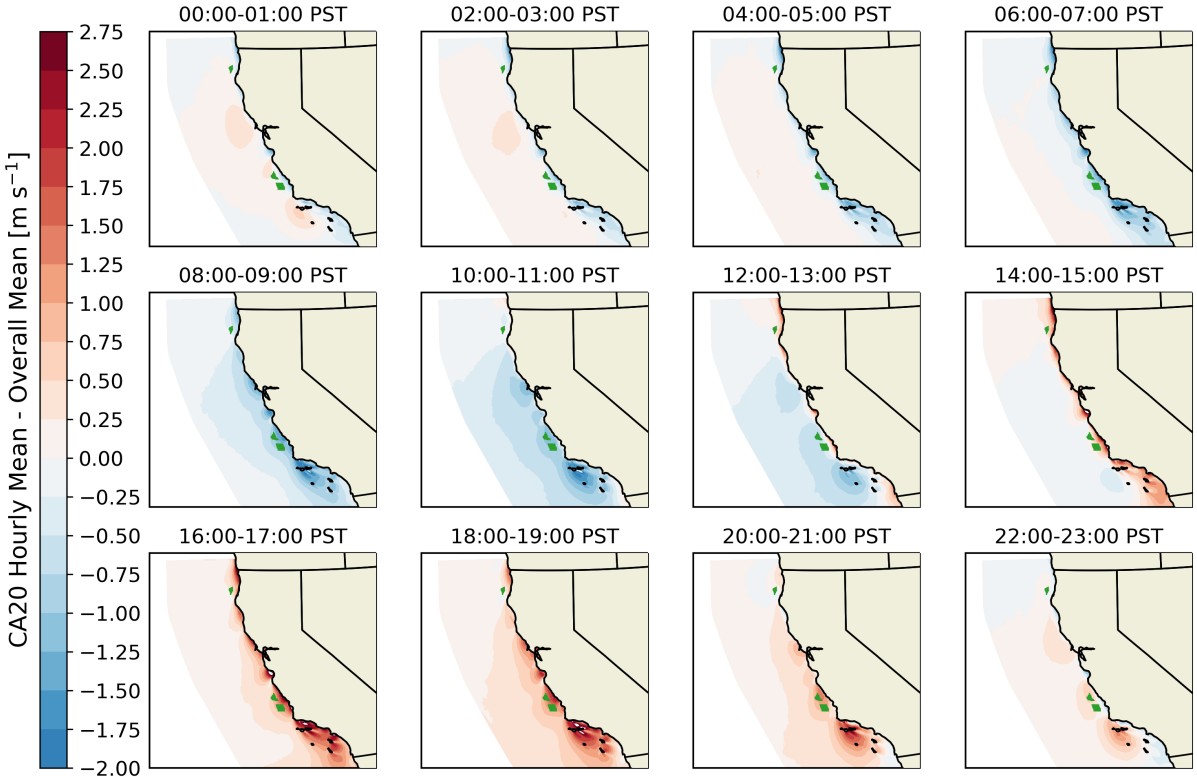

**Figure 4.** Hourly CA20 mean 100-m winds, with reference to the overall CA20 mean. Every other hour is shown for concision.

Of the three timescales, annual-averaged winds showed the smallest period-to-period variability. During the 20-year period, annually averaged CA20 winds deviated between -1.25 to +1.25 m s$^{-1}$ from the overall average (Fig. B1). Annually averaged WIND Toolkit winds showed slightly weaker variability, deviating between -1.0 to +1.0 m s$^{-1}$ from the overall WIND Toolkit mean (Fig. B2).

Monthly averaged winds show the greatest period-to-period variability, ranging from -3.0 to +5.5 m s$^{-1}$ of the overall mean. Throughout the year, the strongest winds within both CA20 (Fig. 3) and the WIND Toolkit (Fig. B3) occur in May, June, and July. The remaining months show substantially smaller deviations from the overall mean.

Variability emerges throughout the day. Hourly averaged winds show strong variability in Southern California for both CA20 (Fig. 4) and the WIND Toolkit (Fig. B4)), ranging from -2.0 to +2.75 m s$^{-1}$ of the overall mean. In all regions, the strongest winds appear in the evening; therefore, offshore wind energy production in this region could help mitigate the California "duck curve," i.e., the decrease in the solar resource in the evening just as demand is increasing (Denholm et al., 2015).

The temporal analysis here agrees with the findings of Wang et al. (2019). That study analyzed WIND Toolkit winds at 125 m and found that offshore wind farm power production peaks during spring and summer as well as during the evening hours



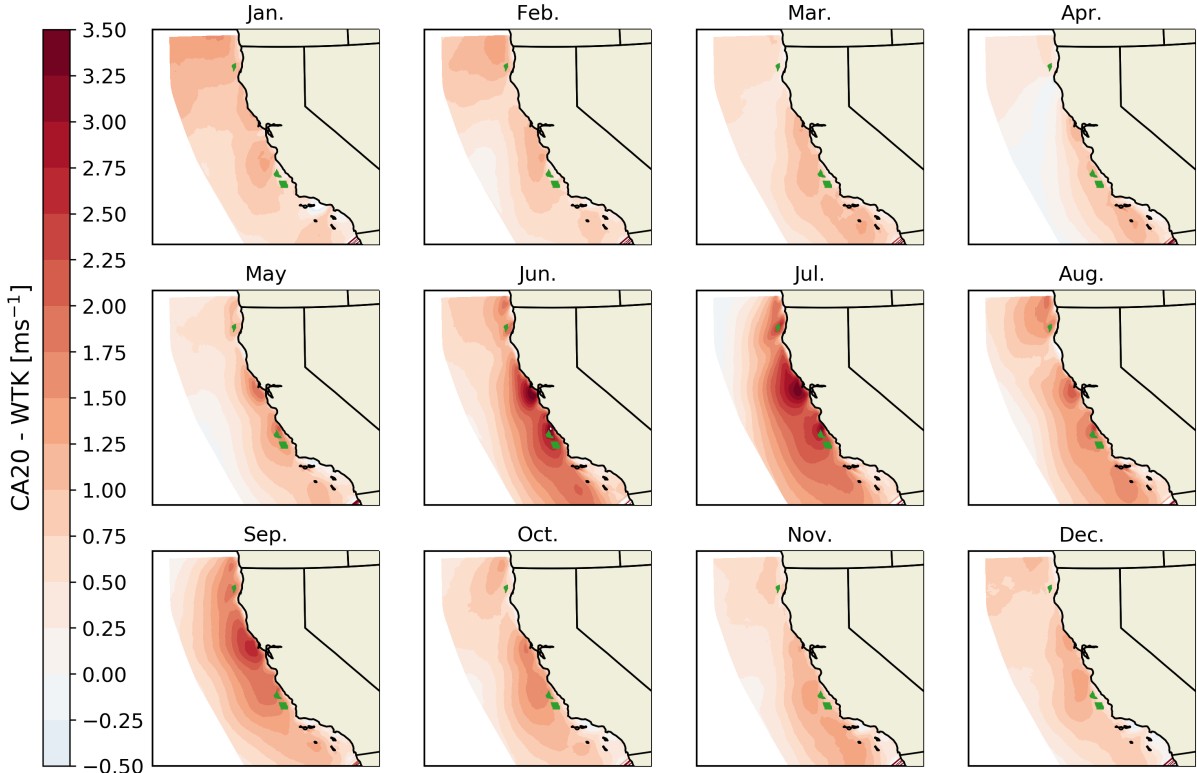

**Figure 5.** Difference in monthly-averaged 100-m winds between CA20 and the WIND Toolkit.

in Central California. Additionally, the study examined power demand in California, finding that offshore wind aligns with demand better than photovoltaics (PV) or onshore wind production.

### 3.3 Temporal Differences Between CA20 and the WIND Toolkit

To better understand the large differences in the overall wind speed between CA20 and the WIND Toolkit, we calculate the difference of yearly, monthly, and hourly averaged winds between the two data sets. The largest variability in differences appear in monthly timescales (Fig. 5). During June and July, CA20 shows monthly mean wind speeds 3.5 m s$^{-1}$ stronger than the WIND Toolkit in Central California. These large wind speeds correspond with the period of strongest winds because monthly mean wind speeds exceed 17 m s$^{-1}$ in June and July. The smallest coastal differences occur in March and April, with
differences of 1.25 m s$^{-1}$ or less. Accordingly, March and April mean winds correspond to a period of relatively calm winds.

The wind speed difference between CA20 and the WIND Toolkit shows smaller variability on hourly averaged and annually averaged scales (Figs. B5 and B6). Maps of wind speed differences show a fairly persistent pattern of large residuals of approximately 2 m s$^{-1}$ in Central California across different hourly and yearly bins. This weak variability on these two



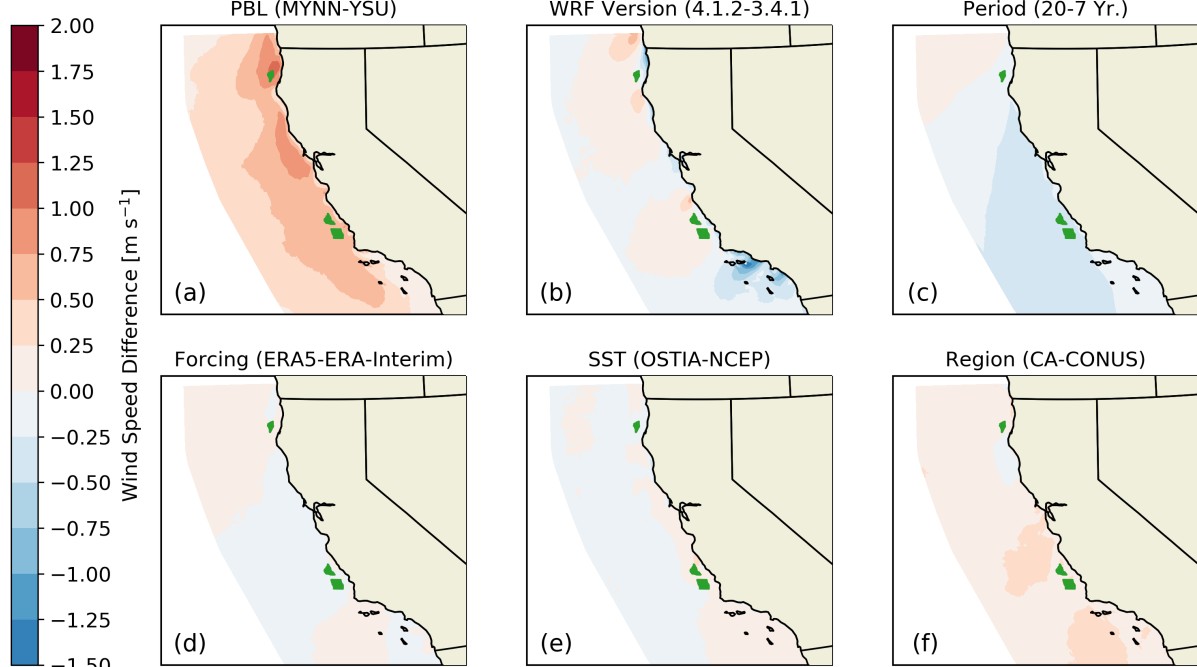

**Figure 6.** Influence of PBL scheme, WRF version, simulation period, reanalysis product, and SST product on CA20 2017 hub-height mean winds.

timescales suggests that the large summer month residuals drive the large overall wind speed difference between CA20 and the

WIND Toolkit.

### 3.4 Sensitivity Analysis

To further investigate the difference in hub-height winds between CA20 and the WIND Toolkit, we conduct a modeling sensitivity analysis. The substantial differences in wind resource between CA20 and the WIND Toolkit stem from differences regarding modeling decisions for both data sets. We estimate the impact of six factors—PBL scheme, WRF version, modeling

duration, reanalysis product, SST product, and domain size—by simulating 1-year data sets in 2017 (Table 2) with options that were used in the WIND Toolkit and comparing them to those of CA20 in 2017.

#### 3.4.1 PBL Scheme

PBL schemes parameterize the effects of turbulence within the atmospheric boundary layer. Hub-height winds from NWP models have been shown to be sensitive to PBL scheme choice (Yang et al., 2017; Berg et al., 2019; Hahmann et al., 2020),

although one study found that in the Baltic Sea, YSU showed negligible differences when compared with the Mellor-Yamada-Janjić (MYJ) scheme (Hahmann et al., 2015). The WIND Toolkit employed the YSU scheme and CA20 used the MYNN





|  | Bulk Ri Range |
| --- | --- |
| Very unstable | (-∞, -0.1) |
| Moderately unstable | (-0.1, -0.025) |
| Mildly unstable | (-0.025, 0) |
| Mildly stable | (0, 0.025) |
| Moderately stable | (0.025, 0.1) |
| Very stable | (0.1, ∞) |

**Table 3.** Bulk Richardson ranges for each stability condition.

scheme (a scheme that shares characteristics with MYJ). Both YSU and MYNN are common options used for wind resource assessments today, but they have substantial differences. For example MYNN is a local scheme, in which fluxes are calculated using only the fields at neighboring vertical levels, whereas YSU is a nonlocal scheme (Cohen et al., 2015). Additionally,

WRF's MYNN parameterization has been updated within recent years as part of the first and second Wind Forecasting Improvement Projects (WFIP1 and WFIP2) (Olson et al., 2019b). Furthermore, in this domain MYNN outperformed YSU relative to observations from 16 NDBC buoys during the pre-production phase of CA20 development (Optis et al., 2020). As such, MYNN was chosen as the PBL scheme used to generate CA20.

We compare CA20 2017 mean wind speeds with those from an otherwise identical simulation that used YSU (Fig. 6a).

Similar to the overall difference between CA20 and the WIND Toolkit, we see larger wind speeds nearly everywhere when using MYNN. The largest PBL-induced differences are on the order of 1 m s$^{-1}$, and they occur in Northern and Central California. In contrast, a sensitivity study for the New European Wind Atlas (NEWA) data set found little difference in winds between YSU and MYNN within a European domain (Witha et al., 2019). As it turns out, the degree of atmospheric stability in the OCS relative to Europe accounts for this discrepancy.

We further investigate PBL-induced differences by examining winds across different atmospheric stabilities. We diagnose stability via the bulk Richardson number in the lowest 200 m (Optis et al., 2016):

$$Ri_B = \frac{g}{0.5\left(\theta_{200} + \theta_2\right)} \frac{z_{200}\left(\theta_{200} - \theta_2\right)}{U_{200}^2} \tag{1}$$

where $g$ is gravitational acceleration, $\theta$ is potential temperature, $z$ is height, and $U$ is wind speed. We employ bins similar to Kalverla et al. (2020), chosen for an offshore environment (Table 3). We calculate the percentage frequency of each stability

across space (Fig. 7). Away from the coast, weakly unstable and weakly stable conditions appear most frequently. Near the coast, very stable conditions dominate, driven by strong coastal upwelling bringing cold water from deep depths to the surface (Huyer, 1983). MYNN shows very stable conditions 10 percentage points more frequently than YSU does in this coastal region (Fig. B7).

These differences in the frequency of stability drive differences in wind speeds. The average winds for each stability condi-

tion (Fig. 8) differ for the two PBL schemes, with the largest difference during moderately stable and weakly stable regimes. In



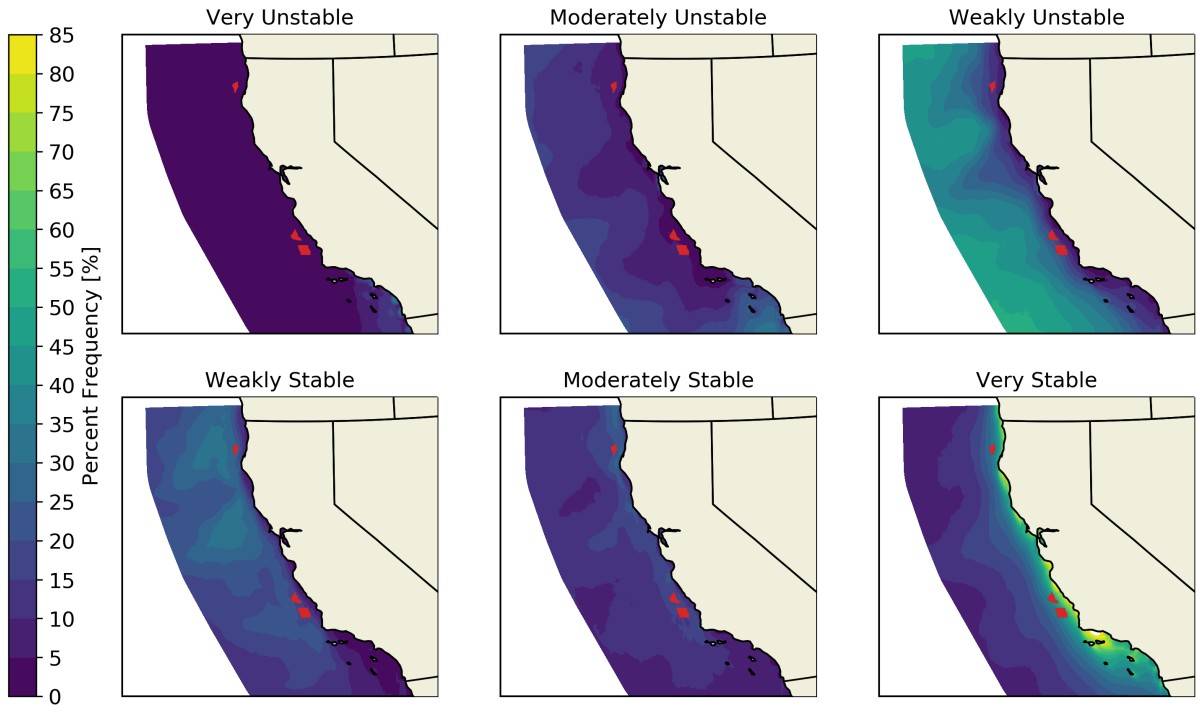

**Figure 7.** Percentage frequency in each stability for MYNN.

moderately stable conditions, MYNN predicts annually averaged winds that are up to 3.5 m s$^{-1}$ stronger than those of YSU. The strongest differences appear in coastal regions in central and northern regions. In contrast, YSU tends to show stronger winds in unstable conditions, in which annually averaged winds are up to 1 m s$^{-1}$ stronger than in MYNN. This behavior corroborates an observation during a NEWA sensitivity analysis in Hahmann et al. (2020), which found that YSU showed slightly

stronger winds than MYNN over waters dominated by unstable stratification; thus, MYNN's tendency to predict stronger winds than YSU occurs in part because MYNN predicts stable conditions more frequently, and these stable conditions have stronger winds.

### 3.4.2 WRF Version

Although the version of WRF leads to some differences in the winds, these differences are smaller than those arising from

the PBL scheme. The WIND Toolkit was generated in 2013 using WRF 3.4.1. Since then, WRF has received many updates, ranging from major (e.g., a shift in vertical coordinate system to a hybrid sigma-pressure scheme) to minor (e.g., bug fixes in microphysics schemes). In this time, the code for the MYNN PBL scheme has also been updated (e.g., Olson et al., 2019a). We compare mean 2017 winds for WRF Version 4.1.2 to those from 3.4.1 (Fig. 6b). We run both simulations with the YSU PBL scheme to isolate the effect of the WRF version from the effect of the MYNN evolution. As shown in Fig. 6b, the WRF

version impacts wind speeds. A handful of small regions show wind speed increases up to 0.75 m s$^{-1}$. Additionally, Southern





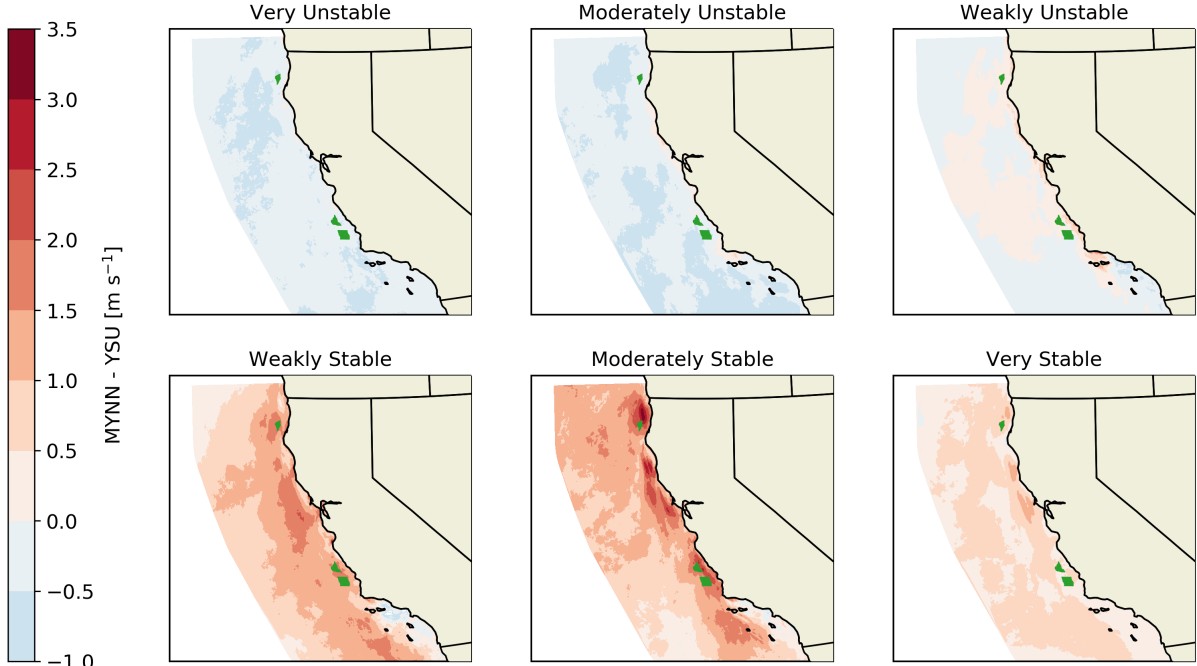

**Figure 8.** The average wind speed difference between MYNN and YSU for each stability bin.

California shows a wind speed reduction that is as strong as 1.5 m s$^{-1}$. These differences are consequential but smaller than the differences due to the PBL scheme.

### 3.4.3 Modeling Duration

The duration of the data set does not explain the differences between CA20 and the WIND Toolkit. Wind resource assessments often aim to represent long-term behavior of winds. If an NWP-based assessment uses a short modeling window, the assessment is susceptible to anomalies (e.g., well below average wind speeds in a given year) driven by climate variability that can bias hub-height winds. Lee et al. (2018) suggest that at least 10 years of data should be used for a wind resource assessment. We examine the impact of the simulation duration by comparing overall mean winds from CA20 to mean winds from CA20 for 2007–2013, the same period used for the WIND Toolkit (Fig. 6c); however, the 20-year window predicts slightly weaker winds (0.5 m s$^{-1}$) than the 7-year window across much of the region. The WIND Toolkit modeling duration was apparently coincident with a period of stronger large-scale forcing in this region.

### 3.4.4 Reanalysis Product and SST Product

Neither the reanalysis product nor the SST product used for forcing these simulations explain the differences between CA20 and the WIND Toolkit. Reanalysis products provide boundary conditions for NWP simulations, and they additionally interact



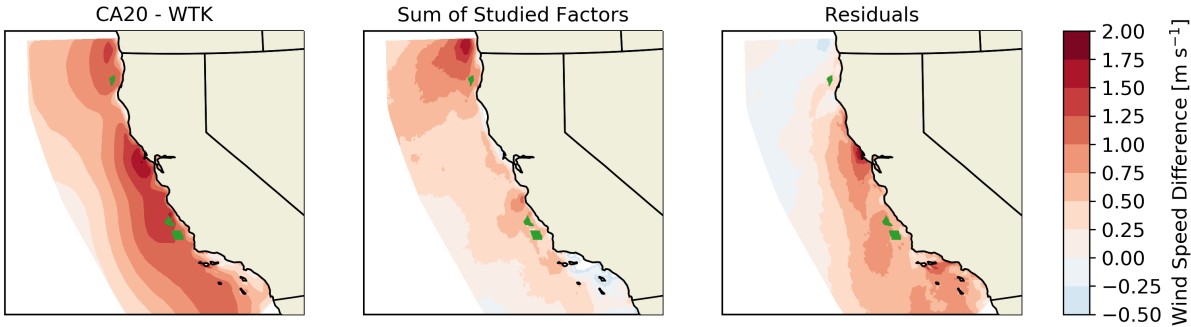

**Figure 9.** Difference in hub-height mean winds between CA20 and the WIND Toolkit, total influence of studied factors between CA20 and the WIND Toolkit, and the residual between maps.

with simulations if nudging is employed. The WIND Toolkit used the ERA-Interim reanalysis product, whereas CA20 used ERA5. The updated product contains higher horizontal resolution (31 km vs. 79 km), higher temporal resolution (1 hour vs. 6 hour), and new input observations that improve the product. We compare CA20 2017 hub-height mean winds to those from a simulation forced by ERA-Interim (Fig. 6d). Wind speed differences are relatively minor, ranging from -0.25 m s$^{-1}$ to 0.25 m s$^{-1}$.

Similarly, we compare the effect of SST products. These provide SST boundary conditions, and they could be included in reanalysis products or they could be stand-alone products. We compare CA20 2017 hub-height winds to those from a simulation forced with the NCEP SST product employed in the WIND Toolkit (Fig. 6e). As with the reanalysis product, we find only minor differences, from -0.25 m s$^{-1}$ to 0.25 m s$^{-1}$.

### 3.4.5 Domain

Finally, part of the wind speed difference between CA20 and the WIND Toolkit likely stems from differences in domain size between the two data sets. Hub-height winds can be sensitive to domain size (Witha et al., 2019; Hahmann et al., 2020). CA20 and the WIND Toolkit have significantly different domains—the WIND Toolkit covers the contiguous United States, whereas CA20 is limited to the state of California. To examine the effect of domain size, we compare 2017 CA20 winds to 2017 winds from an update to the WIND Toolkit that is currently in development (Fig. 6f). This update uses the same physics and forcing

products as CA20. We find that the smaller, California-exclusive domain produces winds that are slightly stronger, from -0.25 m s$^{-1}$ to 0.50 m s$^{-1}$, across most of the coast than the contiguous U.S. domain; thus, it is likely that part of the wind speed difference between CA20 and the WIND Toolkit stems from the different domains.

### 3.4.6 Overall Explained Change

We attempt to account for the cumulative effect of these differences by summing the contributions from each component in

the sensitivity analysis. (Fig. 9b). This simple summation does not account for the fact that NWP models are highly nonlinear,





so a linear summation cannot include these nonlinear interactions. For example, the combined impact of both higher vertical resolution and a different PBL scheme might be stronger than the sum of the individual contribution from each. Nonetheless, in this sensitivity analysis and consistent with other analyses, we find that both the PBL scheme and the WRF version significantly impact modeled wind resource. The residuals between the overall data set difference map (Fig. 9) highlight regions where

these nonlinear interactions are significant. In the end, the shift in wind speed (Fig. 9a) is fairly well accounted for in Northern California (Fig. 9c); however residuals remain large in Central and Southern California, at 2 m s$^{-1}$. At the Call Areas, these unexplained differences are 0.17 m s$^{-1}$ at Humboldt, 0.99 m s$^{-1}$ at Morro Bay, and 1.11 m s$^{-1}$ at Diablo Canyon.

## 4 Call Area Analysis

Having discussed broad differences across the entire OCS between both data sets, we now provide a refined wind resource

analysis at the centroid of each of the three California Call Areas. By focusing on winds at centroids, we parallel the type of measurements that would be obtained from a single lidar during a measurement campaign. We first characterize wind speed distributions and quantify power production. We then follow by assessing atmospheric stability, wind shear, directional veer, and wind droughts. A scaled-down 10-MW version (Beiter et al., 2020) of the International Energy Agency (IEA) 15-MW offshore reference wind turbine (Gaertner et al., 2020) is used for key parameters such as cut-in and cut-out speed. As a result,

in this section we take "hub-height winds" at 120 m instead of 100 m as in the prior section.

### 4.1 Wind Speed Distributions

Although data set-averaged wind speeds are useful to provide a high-level view of wind resource, it is important to characterize the distribution of hub-height wind speeds through time to estimate characteristics such as power output and structural loads (Morgan et al., 2011; Petković et al., 2014; Shu et al., 2016). Distributions provide a sense of how frequently winds appear

below the cut-in speed, at rated speed, or above the cut-out speed.

At all three sites, the relative frequency distribution of hub-height wind speeds bears resemblance to the Weibull distribution (Fig. 10). This simple distribution is commonly used to characterize the spread of wind speeds, and hub-height winds have been observed to follow this distribution in an offshore environment (Shu et al., 2016). Winds most frequently appear between the cut-in speed of 3 m s$^{-1}$ and the rated speed of 10.6 m s$^{-1}$, depending on the model and the Call Area. The WIND Toolkit

produces winds weaker than the rated wind speed more frequently than CA20, and its relative frequency distribution shows a sharper drop off in Region III of the power curve. Conversely, CA20 predicts winds stronger than the rated speed more frequently than the WIND Toolkit; thus, based on these curves, we expect CA20 to predict higher power output than the WIND Toolkit.

### 4.2 Power Production

Having characterized wind speed distributions at the three Call Areas, we conduct an idealized power production study. As discussed, wind resource varies through time, and therefore it is key to understand how power production varies across different

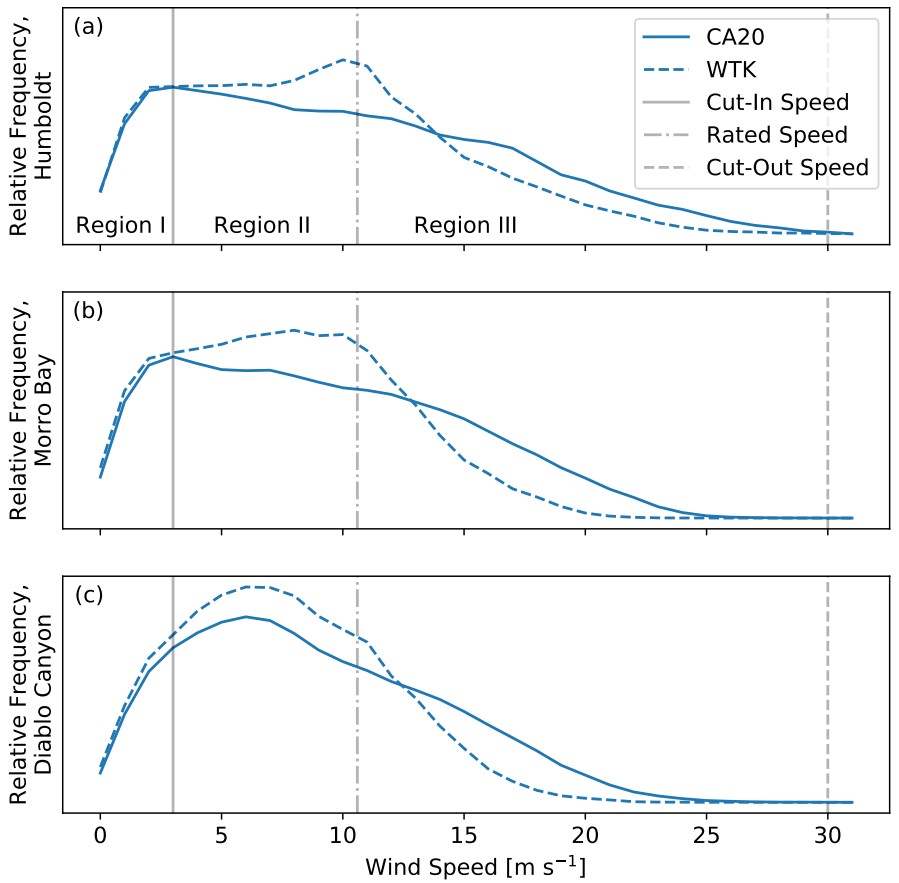

**Figure 10.** Hub-height wind speed distributions at (a) Humboldt, (b) Morro Bay, and (c) Diablo Canyon.

timescales (King et al., 2014). We estimate power production by processing modeled wind speeds with the scaled-down IEA 10-MW power curve.

The capacity factor is calculated for each month by comparing estimated power output to the maximum possible power

output (Fig. 11). As expected, the three sites exhibit significant seasonality. Year-round, capacity factors range from 0.3 to 0.7. Power production peaks in late spring and in early summer, although the timing varies slightly by site. Humboldt shows a maximum capacity factor in July (∼0.7), whereas Morro Bay and Diablo Canyon show maximums in May and June (∼0.65). Morro Bay and Diablo Canyon drop to their lowest capacity factors in winter (0.3–0.4). Humboldt also shows a decrease in power output during this period, although not as low (∼0.5).

CA20 shows a larger power output than the WIND Toolkit nearly consistently. The largest differences in capacity factor occur in July, where CA20 shows a capacity factor that is 0.13 larger in both Morro Bay and Diablo Canyon. In general, CA20 and the WIND Toolkit show larger capacity factor discrepancies at Morro Bay and Diablo Canyon rather than in Humboldt. The models agree at times—both models show nearly identical performance across all three sites in April.



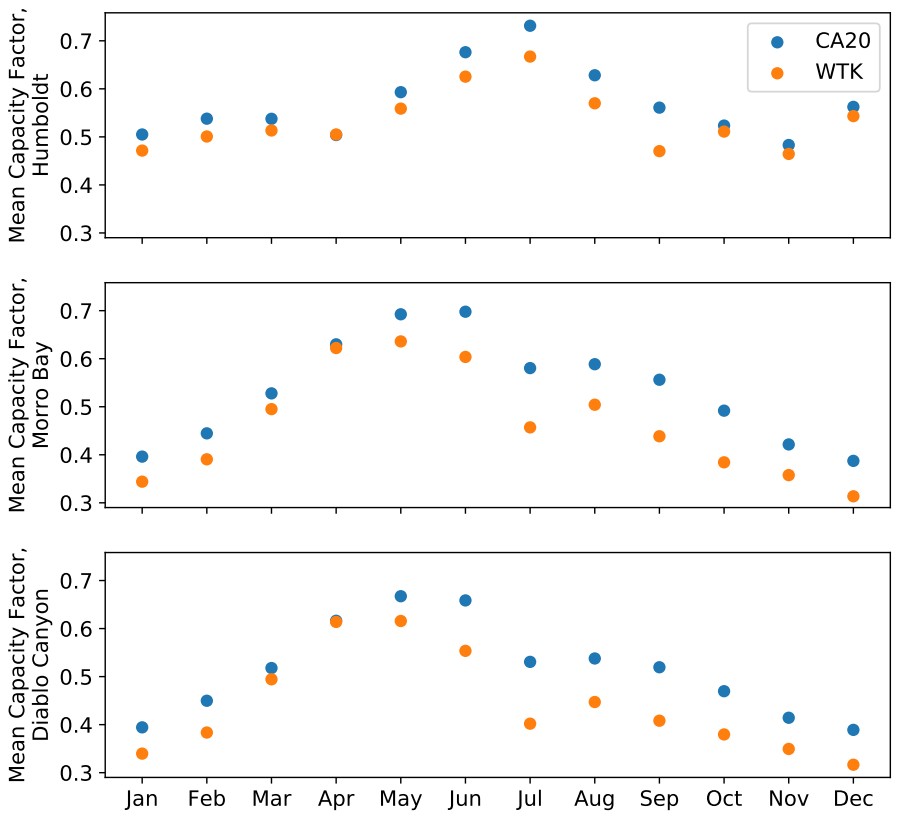

**Figure 11.** Average monthly capacity factors at (a) Humboldt, (b) Morro Bay, and (c) Diablo Canyon.

Having quantified power output under idealized inputs (using only wind speeds and power curves), we turn to characterizing
other qualities that have been shown to impact power production.

## 4.3   Stability Analysis

As discussed in Sect. 3.4.1, atmospheric stability directly impacts mean wind speeds. Furthermore, atmospheric stability modifies turbulence and turbulence intensity, thereby impacting power production and structural stresses (Wharton and Lundquist, 2012; Sathe et al., 2013; St. Martin et al., 2016). We characterize atmospheric stability at the centroids of the Call Areas
(Fig. 12). Within both data sets, near-neutral conditions dominate at all three sites. These values correspond to values in Fig. 7 because the Call Areas sit between the waters directly adjacent to the coastline (that are predominantly very stable) and the deeper waters (that are predominantly weakly unstable). At these locations, CA20 shows a greater frequency of stable conditions, whereas the WIND Toolkit shows a greater frequency of unstable conditions.



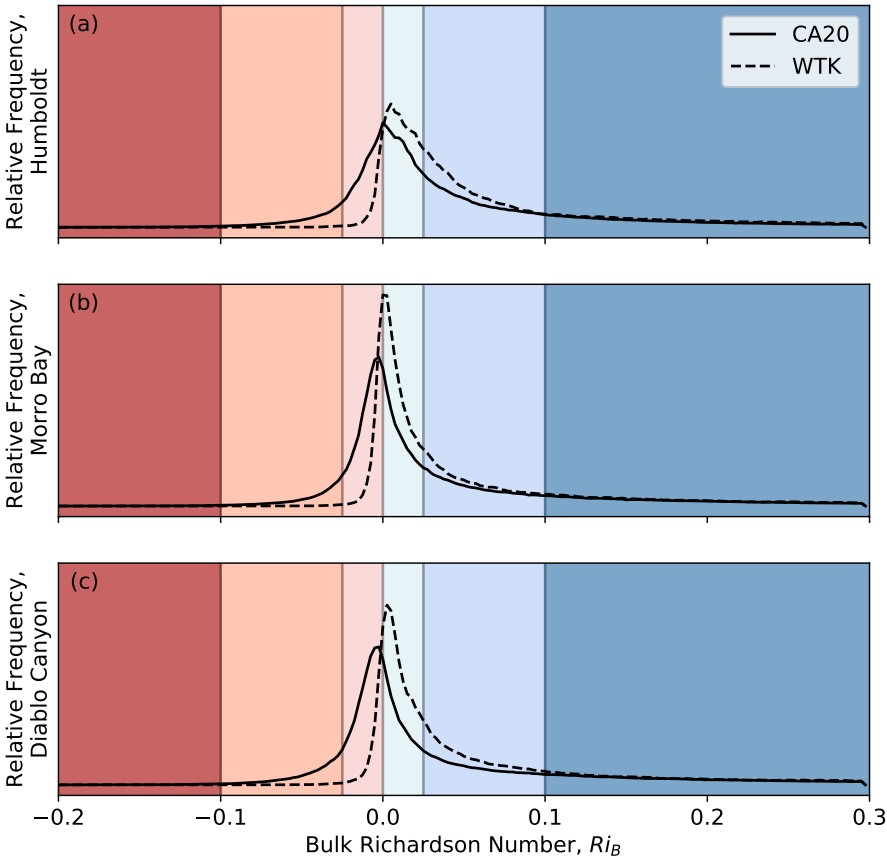

**Figure 12.** Bulk Richardson number distributions at (a) Humboldt, (b) Morro Bay, and (c) Diablo Canyon. Regions are shaded corresponding to Table 3. From left to right, colors correspond to very unstable, unstable, mildly unstable, mildly stable, stable, and very stable.

## 4.4 Wind Shear and Veer

### 4.4.1 Definitions, Observed Values, and Impact on Power Production

When moving from the bottom to the top of the rotor disk, wind speed and direction change with height. These changes are referred to as shear and veer, respectively. They stem from factors such as surface friction, vertical gradients in geostrophic winds, and the Coriolis effect (Walter et al., 2009), and they vary with atmospheric stability (Wharton and Lundquist, 2012).

Wind shear is often characterized using the dimensionless wind shear exponent, $\alpha$, for wind energy applications, defined as:

$$\alpha = \frac{\ln \frac{WSPD_2}{WSPD_1}}{\ln \frac{z_2}{z_1}},$$ (2)





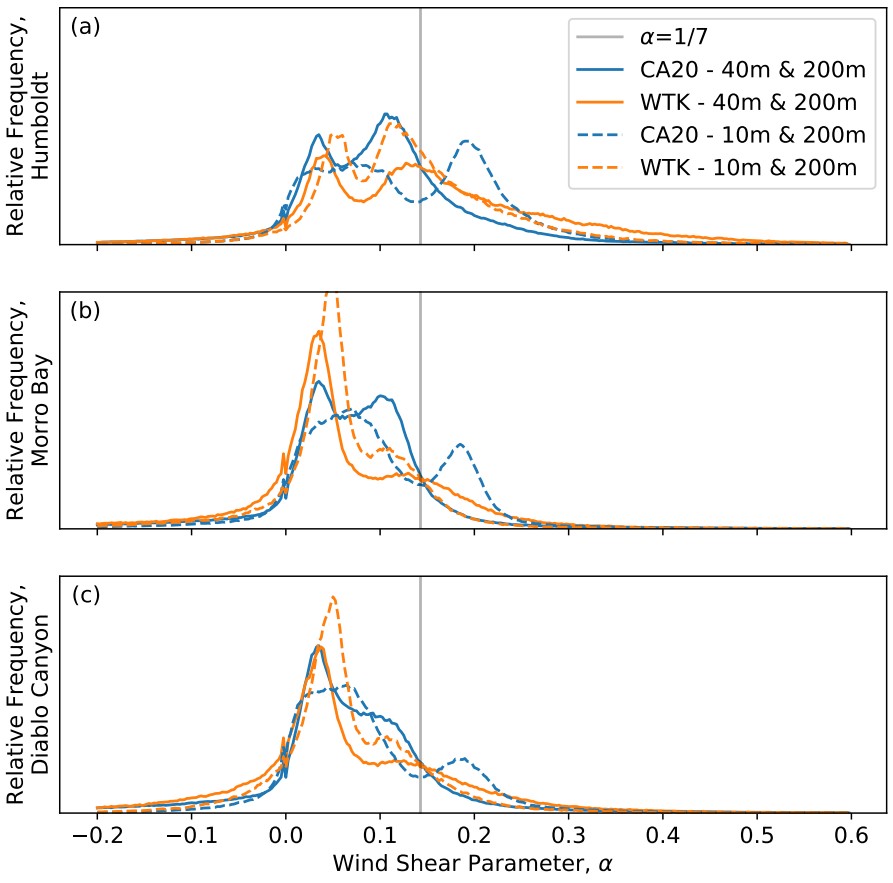

**Figure 13.** Wind shear distributions at (a) Humboldt, (b) Morro Bay, and (c) Diablo Canyon. The noisy behavior near $\alpha = 0$ is an artefact of numerical calculations.

where $z_2 = 200$ m is the upper height in all calculations. We test the sensitivity of shear to the lower wind height, taking $z_1$ as 10 m and 40 m.

To extrapolate wind speed measurements upward and downward to different elevations, a constant value of $\alpha = 0.14$ has often been employed in wind resource assessments, although this value has been established with neutral conditions in mind (Walter et al., 2009; Shu et al., 2016). Using lidar measurements from an offshore platform near Hong Kong, Shu et al. (2016) calculated shear coefficients across different height intervals $(z_1, z_2)$, observing that $\alpha \approx 0.10$, with fairly small variability in the mean ($\sim 0.01$) across different intervals.

We calculate veer:

$$veer = \frac{\theta_2 - \theta_1}{z_2 - z_1} \tag{3}$$

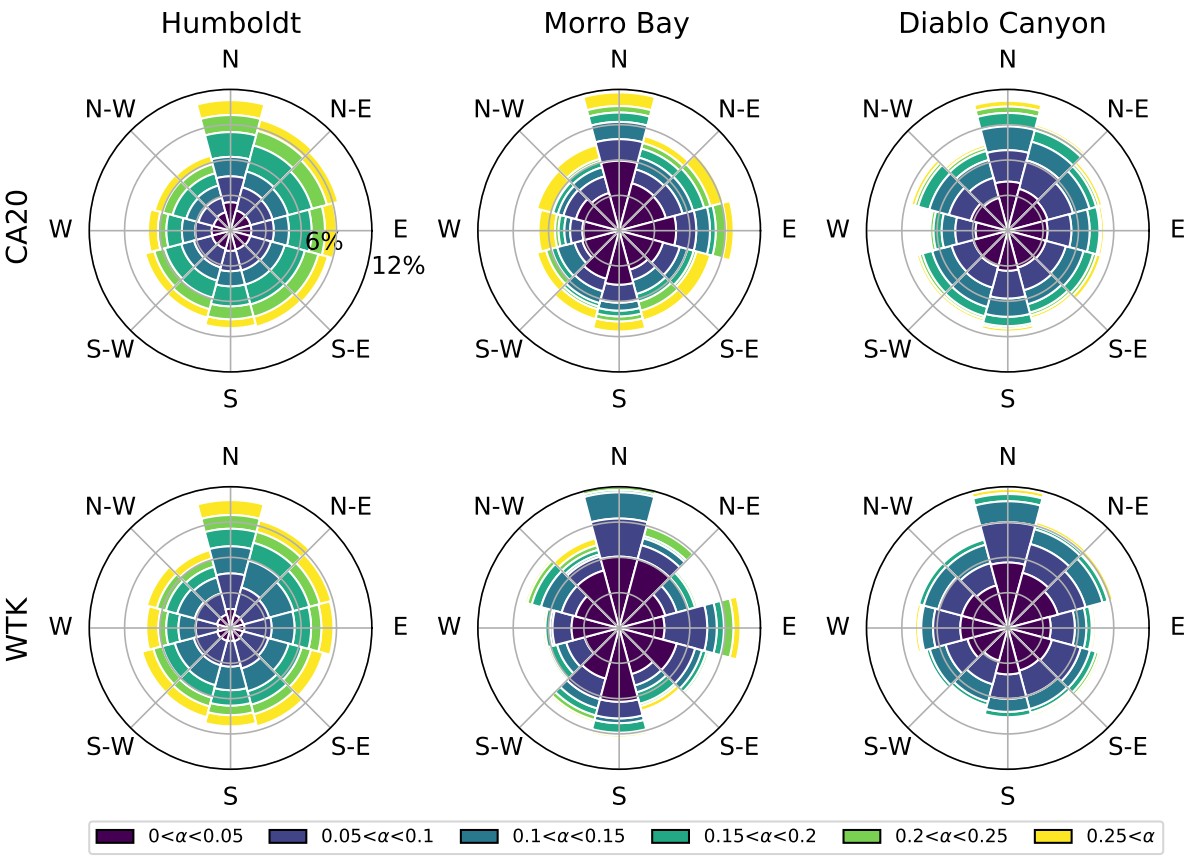

**Figure 14.** Shear (calculated between 10 and 200 m) as a function of incoming wind direction. Radial bands are separated 3% apart.

in units of $^\circ$ m$^{-1}$ using the same heights as for the shear calculation. For both shear and veer calculations, we neglect winds weaker than the cut-in speed and stronger than the cut-out speed.

Recent observational studies have measured veer in offshore environments. Bodini et al. (2019) and Bodini et al. (2020) measured winds off the coast of Massachusetts and observed that the average wind veer was twice as strong in the summer as it was in winter (where the magnitude of veer was 0.10 vs. 0.05 $^\circ$ m$^{-1}$), and summertime conditions occasionally produced veer

greater than 0.3 $^\circ$ m$^{-1}$. Shu et al. (2020) measured veer on hourly intervals from an offshore platform in Hong Kong. They also found a seasonality to their veer, but veer was strongest in winter there. Additionally, wind veer decreased as wind speeds increased. When inflow winds came from the open sea, average veer was approximately 0.005 $^\circ$ m$^{-1}$ for winds from 14–17 m s$^{-1}$ but 0.018 $^\circ$ m$^{-1}$ for winds from 5–8 m s$^{-1}$. When inflow winds came from hilly terrain, average veer was approximately 0.027 $^\circ$ m$^{-1}$ for winds 14–17 m s$^{-1}$ and 0.056 $^\circ$ m$^{-1}$ for winds 5–8 m s$^{-1}$.

Crucially, wind shear and veer have been shown to impact wind turbine power production through structural modeling studies (Walter et al., 2009; Wagner et al., 2010) as well as observation-based studies (Wharton and Lundquist, 2012; Vanderwende

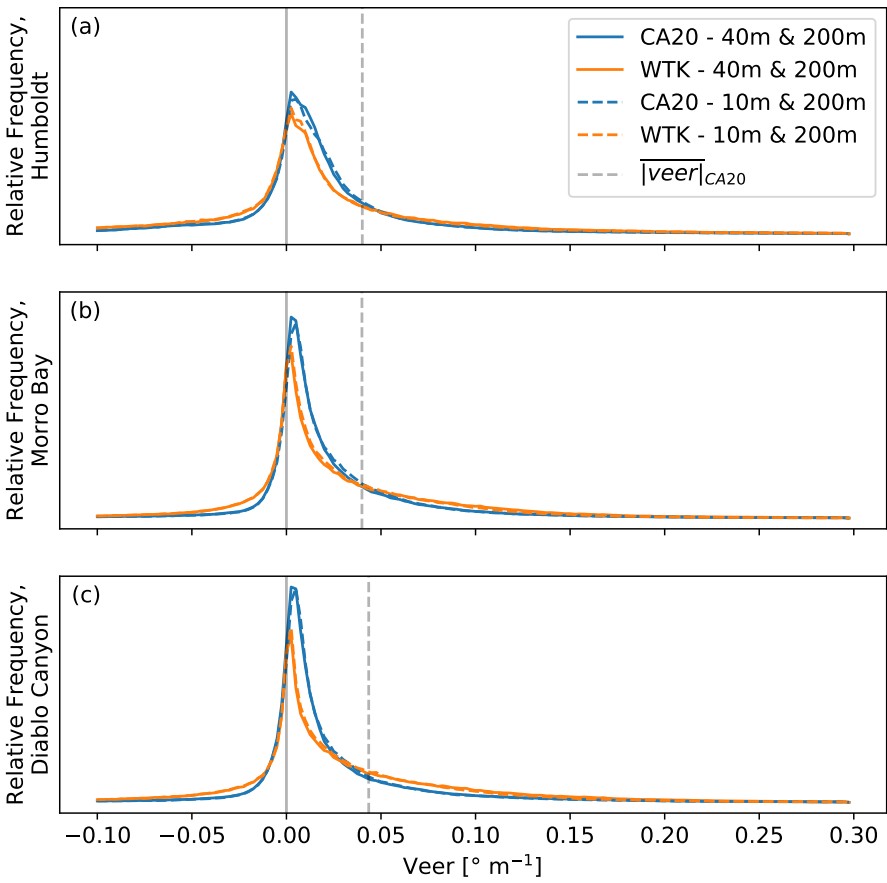

**Figure 15.** Wind veer distribution at (a) Humboldt, (b) Morro Bay, and (c) Diablo Canyon. The solid gray line demarcates positive from negative values of veer. The average magnitude of veer is shown as a gray dashed line.

and Lundquist, 2012; Optis and Perr-Sauer, 2019; Sanchez Gomez and Lundquist, 2020b, a). The results from these studies show that although the relationship between shear, veer, and power output is complicated, shear and veer can have significant impacts on power production. For example, Sanchez Gomez and Lundquist (2020b) found that normalized power production

can reduce normalized power production to 75% (for $\alpha \approx -0.05$ and a veer of approximately $0.15\,°\,\mathrm{m}^{-1}$ in this study) or boost normalized power production to 108% (for $\alpha \approx 0.75$ and a veer of approximately $0.05\,°\,\mathrm{m}^{-1}$ in this study).

### 4.4.2 Modeled Shear

In contrast to the Weibull distribution of hub-height wind speeds, $\alpha$ exhibits a bimodal distribution at the Call Areas. Distributions at all sites show a peak near a weaker value of $\alpha = 0.04$. A secondary peak appears at larger values of $\alpha$, although the

particular value of this secondary peak depends on the data set and range of heights used for the calculation. Hahmann et al. (2015) also found double peaks of shear distributions in an offshore environment, and they found that this behavior stemmed





from upstream winds that originated on land versus offshore; however, at these three Call Areas, the distribution of shear does not significantly change between upstream-land (east) and upstream-ocean. For example, at Humboldt, easterly winds occur more often than westerly winds; however, the distributions of shear from these directions are nonetheless similar.

The height interval that is used to calculate shear can be important. On one hand, the secondary peak from both WIND Toolkit distributions (10–200m, 40–200m) occurs near $\alpha = 0.11$ at all three sites. In CA20, however, the 10–200-m shear coefficient peak occurs near $\alpha = 0.18$, but the 40–200-m peak occurs near $\alpha = 0.11$; thus, in contrast to the observed distributions of $\alpha$ in Shu et al. (2016), it appears that shear from modeled winds might be significantly sensitive to the heights $z_1$ and $z_2$. The presence of the secondary lobe for CA20 10–200 m suggests the development of very shallow stable layers that have capping
inversions at a height between 10 and 40 m. These shallow stable layers are not predicted in the WIND Toolkit.

### 4.4.3   Modeled Veer

All three sites show a peak frequency of veer near $0\,°\,m^{-1}$. As is typical in the Northern Hemisphere, positive instances of veer appear more frequently than negative instances of veer ("backing"). Veer distributions at Diablo Canyon and Morro Bay show sharper peaks than at Humboldt. Relative to the wind speed distributions and the shear distributions, CA20 and the WIND
Toolkit show relatively small differences in veer. The WIND Toolkit predicts slightly more frequent instances of backing. The average magnitude of veer at all three sites is approximately $0.4\,°\,m^{-1}$. This is larger than the average veer observed in Shu et al. (2020) but smaller than the average veer in Bodini et al. (2019), likely corresponding with the predominantly open-sea inflow at the California Call Areas.

    Taken together, the most frequent conditions at all three call sites are a veer of approximately $0\,°\,m^{-1}$ and $0 \leq \alpha \leq 0.2$. If
the land-based 1.5-MW turbine analysis of Sanchez Gomez and Lundquist (2020b) holds for these Call Areas, then normalized power production will remain approximately 99%–100% of what would be expected; however, it would be fair to assume that in this offshore environment with substantially larger turbines, the normalized power production would shift.

### 4.5   Wind Droughts

During operation, turbines can be subject to prolonged periods with weak winds and minimal power production. During these
intervals, electricity must be supplied to the power grid through alternative sources. These periods are known by many names, such as wind droughts (Katzenstein et al., 2010; Lledó et al., 2018), low-wind-power events (Handschy et al., 2017; Ohlendorf and Schill, 2020), and (when simultaneously joined by dark skies and minimal PV power production) "Dunkelflaute" (Li et al., 2020). Although these studies discuss similar events, each uses a different metric to define a wind drought.

    In this study, we define a wind drought following one "constantly below threshold" metric from Ohlendorf and Schill
(2020). Here, a wind drought is a continuous period where the capacity factor is less than 10%. We calculate the number of wind droughts per year that are at least 6 hours long, choosing this minimum duration threshold arbitrarily (Fig. 16). The distributions of drought between CA20 and the WIND Toolkit agree well with one another, not showing any obvious signs of bias between the two models. The three sites also experience similar amounts of drought.



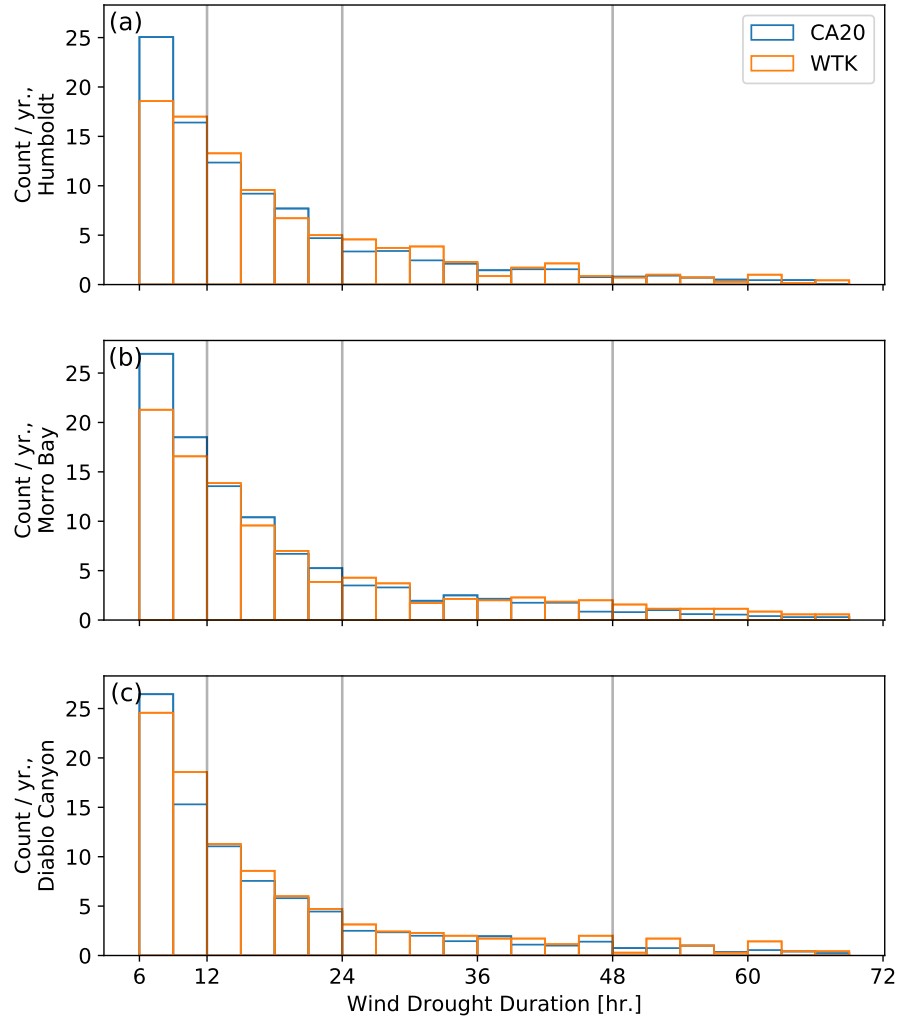

**Figure 16.** Wind drought distribution at (a) Humboldt, (b) Morro Bay, and (c) Diablo Canyon.

Each year, all three sites experience on average approximately 40 droughts that are between 6–12 hours long, 30 droughts
between 12–24 hours long, 16 droughts between 24–48 hours long, 4 droughts between 48–72 hours long, and 2 droughts
longer than 72 hours. Of all the droughts in both data sets, the longest occurred in the WIND Toolkit's Diablo Canyon, lasting
162 hours. Across the three Call Areas, persistent droughts ($\geq$ 24 hours long) did not show an obvious pattern interannually
or during an annual cycle (Fig. 17). When wind droughts were present somewhere along the coast, only one site was impacted
56% of the time, two sites were impacted 36% of the time, and all three sites were impacted 8% of the time. This three-site
drought case represents approximately 290 hours of a typical year; thus, if wind farms are built at all three Call Areas, the grid
will only infrequently experience low power output from all offshore sites.





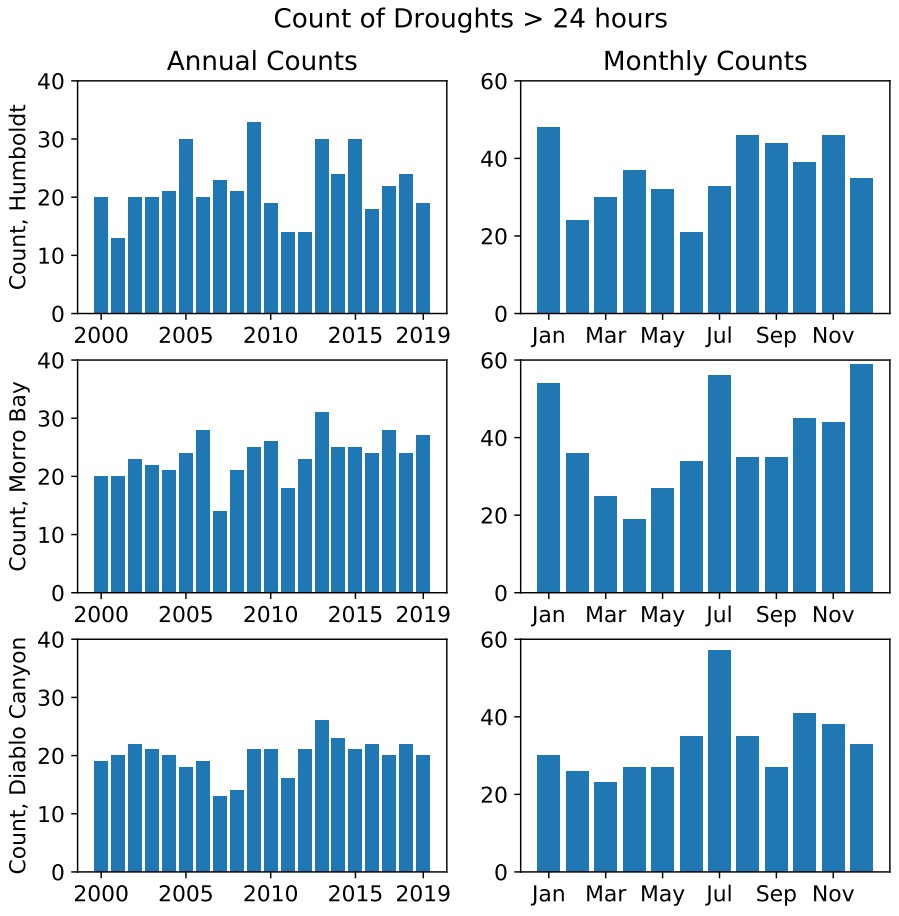

**Figure 17.** Distribution of wind droughts longer than 24 hours from the CA20 data set across years (left) and months (right).

## 5 Conclusions

In this study, we analyzed CA20, a simulated 20-year data set of winds in offshore California. This data set serves as an update to NREL's WIND Toolkit (Draxl et al., 2015) in this region, incorporating advances such as an updated PBL scheme as well as an updated renalysis product for forcing boundary conditions. Overall, CA20 predicts significantly stronger average hub-height winds than the WIND Toolkit, a difference of up to $1.75\,\mathrm{m\,s^{-1}}$ when winds are averaged across entire data sets. This difference is strongest in June and July, where monthly mean winds are $3.5\,\mathrm{m\,s^{-1}}$ stronger in offshore Central California in the new data set. To examine the sources of these differences, we carry out an ensemble of single-year simulations to study factors that likely drive this increased wind resource—PBL scheme, WRF version, period of the simulation, reanalysis product, SST product, and domain size. Of the studied factors, the switch in PBL scheme from YSU to MYNN induces the largest impact. In particular, MYNN tends to predict stronger winds in stably stratified conditions, which frequently occur near the coast.





Through this sensitivity study, we explain a large portion of the wind resource shift in Northern California but not in the other parts of the state. The unexplained wind speed residuals might stem from nonlinear interactions of the studied factors.

Additionally, we conducted a refined wind resource analysis at three Call Areas. Corresponding to the overall stronger winds in CA20, the WIND Toolkit more frequently shows winds within Region II of a 10-MW turbine's power curve, whereas CA20 more frequently shows winds within the rated wind speed region (Region III). Following this, CA20 shows higher capacity factors than the WIND Toolkit for nearly every month of the year. The largest capacity factors (~0.70) are between May and July. Near-neutral atmospheric stability dominates at these Call Areas, with CA20 showing a slight preference for stable conditions and the WIND Toolkit for unstable conditions. CA20 and the WIND Toolkit show similar degrees of veer, with

CA20 showing slightly more cases of near-zero veer. Shear exhibits bimodal behavior, and it is also more sensitive to the choice of Call Area, model data set, and height interval across which shear is calculated. Finally, we quantified the frequency of wind droughts, finding that all three sites experience approximately 40 wind droughts between 6–12 hours long each year and 2 wind droughts that are longer than 72 hours each year. When wind drought conditions are present, they simultaneously occur at all three sites only 8% of the time.

CA20 underscores that offshore wind energy is well positioned to help meet California's energy demands. Offshore wind will have the strongest resource in the summer months as well as during the evenings. This resource is located relatively close to major population centers, which is beneficial for power transmission and distribution. CA20 also showed significantly different wind resource than the previous state-of-the-art data set, highlighting the current rapid pace of NWP model development. Work is currently underway to produce a new 20-year data set that covers the entirety of the contiguous United States. This data set

will provide additional characterization of uncertainty, and it will also include offshore California, thereby providing another set of modeled winds that can be used to assess wind resource in this region. Finally, these differences underscore the crucial role of observations, especially hub-height observations of wind speeds. Future work will compare simulated winds from these data sets to observed winds from offshore lidar.

*Code and data availability.*

The CA20 data set is available at 5-minute and hourly resolution through a Python interface by following instructions at https://github.com/NREL/hsds-examples/blob/master/datasets/WINDToolkit.md. The AWS Registry link to this data set as well as other NREL data sets may be found at https://registry.opendata.aws/nrel-pds-wtk/. The namelists used to generate CA20 can be found at https://doi.org/10.5281/zenodo.45975482280. Furthermore, this link contains post-processed data and scripts used to generate the figures in this manuscript. The WRF model may be accessed at https://github.com/wrf-model/WRF.

**Appendix A: Model-Buoy Comparison**

Under ideal conditions, hub-height wind speed measurements would be used to validate CA20 and the WIND Toolkit in off-shore California. At the time of publication, such measurements are not available for a large time window, but an observational



| Buoy | Bias [m s$^{-1}$] | | | RMSE [m s$^{-1}$] | | |
|---|---|---|---|---|---|---|
| | WTK | CA7 | CA20 | WTK | CA7 | CA20 |
| 46027 | 0.58 | 0.92 | 0.78 | 2.67 | 2.79 | 2.77 |
| 46022 | -0.03 | 0.04 | 0.13 | 1.99 | 2.01 | 2.06 |
| 46014 | 0.11 | 1.09 | 1.08 | 1.91 | 2.46 | 2.47 |
| 46013 | -0.28 | 0.12 | -0.01 | 1.71 | 1.80 | 1.88 |
| 46026 | -0.24 | 0.25 | 0.15 | 1.70 | 1.93 | 1.96 |
| 46012 | -0.49 | 0.13 | 0.25 | 1.61 | 1.66 | 1.73 |
| 46042 | -0.65 | -0.11 | -0.08 | 1.75 | 1.70 | 1.71 |
| 46028 | -0.47 | 0.18 | 0.19 | 1.66 | 1.62 | 1.64 |
| 46011 | 0.13 | 0.55 | 0.46 | 1.69 | 1.84 | 1.80 |
| 46023 | -0.20 | 0.22 | 0.19 | 1.58 | 1.72 | 1.76 |
| 46063 | -0.12 | -0.02 | -0.00 | 1.49 | 1.64 | 1.65 |
| 46054 | -0.24 | -0.25 | -0.49 | 1.79 | 2.00 | 2.02 |
| 46053 | 0.30 | 0.20 | 0.14 | 1.94 | 2.06 | 2.07 |
| 46025 | -0.24 | 0.19 | 0.19 | 1.74 | 1.93 | 1.89 |
| 46069 | -0.38 | 0.04 | 0.06 | 1.63 | 1.71 | 1.63 |
| 46047 | -0.78 | 0.12 | 0.01 | 1.54 | 1.28 | 1.28 |
| 46086 | -0.46 | 0.29 | 0.30 | 1.49 | 1.49 | 1.49 |
| Avg. | **-0.20** | **0.23** | **0.20** | **1.76** | **1.86** | **1.87** |

**Table A1.** Bias and root mean square error (RMSE) for each buoy averaged across the full data sets.

campaign commenced in October 2020 to measure hub-height winds with two lidars—one off the coast of Humboldt County
and the other near Morro Bay. A future follow-up study will compare lidar-measured winds to modeled winds. Nonetheless,
buoy measurements of surface winds can be used as an anchor to the real world, but they miss key features that impact wind
resource aloft, such as low level jets. Buoy measurements have two distinct advantages over the lidar measurements. First,
there are many buoys and they are spread out over a large area. Second, many of the buoys have been measuring for several
years. Here, we evaluate the performance of CA20, CA7, and the WIND Toolkit relative to buoy measurements.

Overall, CA20 and the WIND Toolkit show a similar magnitude bias—CA20 shows an average positive bias (0.20 m
s$^{-1}$), whereas the WIND Toolkit shows an average negative bias (-0.20 m s$^{-1}$). The models also show regional variability
in performance—the WIND Toolkit tends to have smaller bias in the north, whereas CA20 tends to have smaller in the south.
The WIND Toolkit and CA20 have a similar degree of RMSE, 1.76 and 1.87 m s$^{-1}$ respectively, though the WIND Toolkit
has a slightly smaller RMSE. Thus, based on buoy measurements, it is difficult to say that one data set is overall "better" than
the other. These comparisons suggest that CA20 performs better in Northern California whereas the WIND Toolkit performs





better in Southern California. Future comparisons to lidar measurements will measure winds at rotor disk heights, and these

comparisons will provide additional insight on which dataset performs better overall.



**Appendix B: Additional Temporal Analysis Figures**



**Figure B1.** Annually averaged 100-m CA20 winds, with reference to the overall CA20 mean.



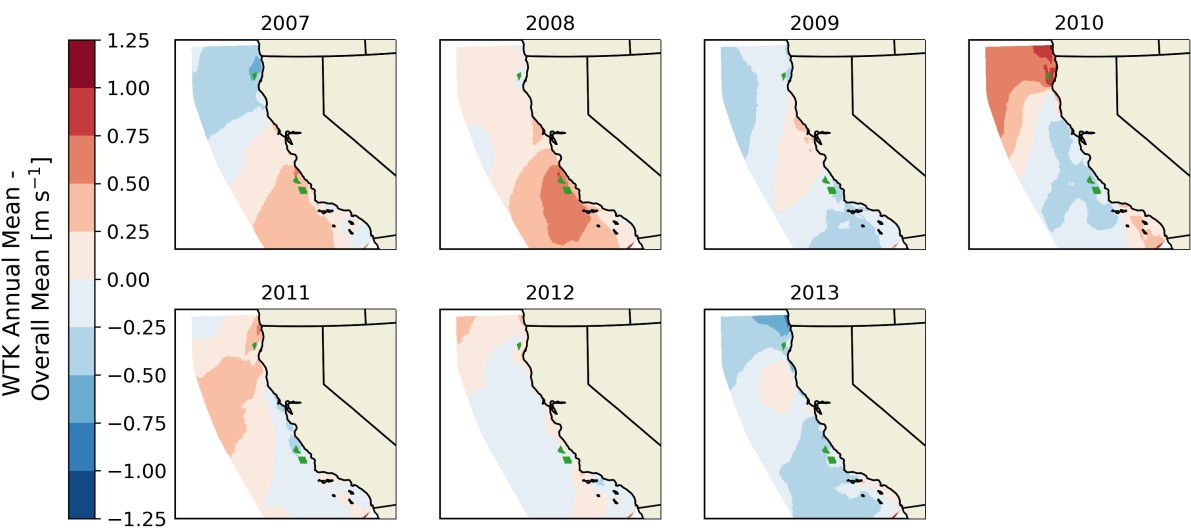

**Figure B2.** Annually averaged 100-m WIND Toolkit winds, with reference to the overall WIND Toolkit mean.

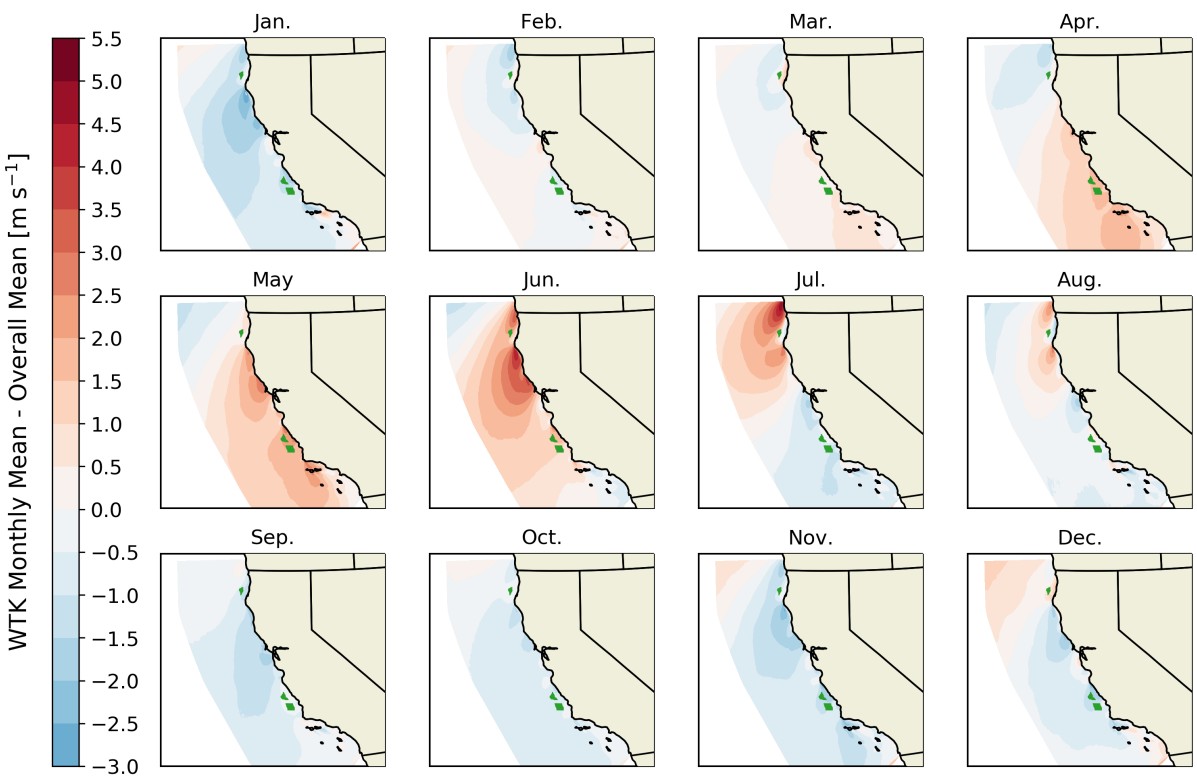

**Figure B3.** Monthly averaged 100-m WIND Toolkit winds, with reference to the overall WIND Toolkit mean.



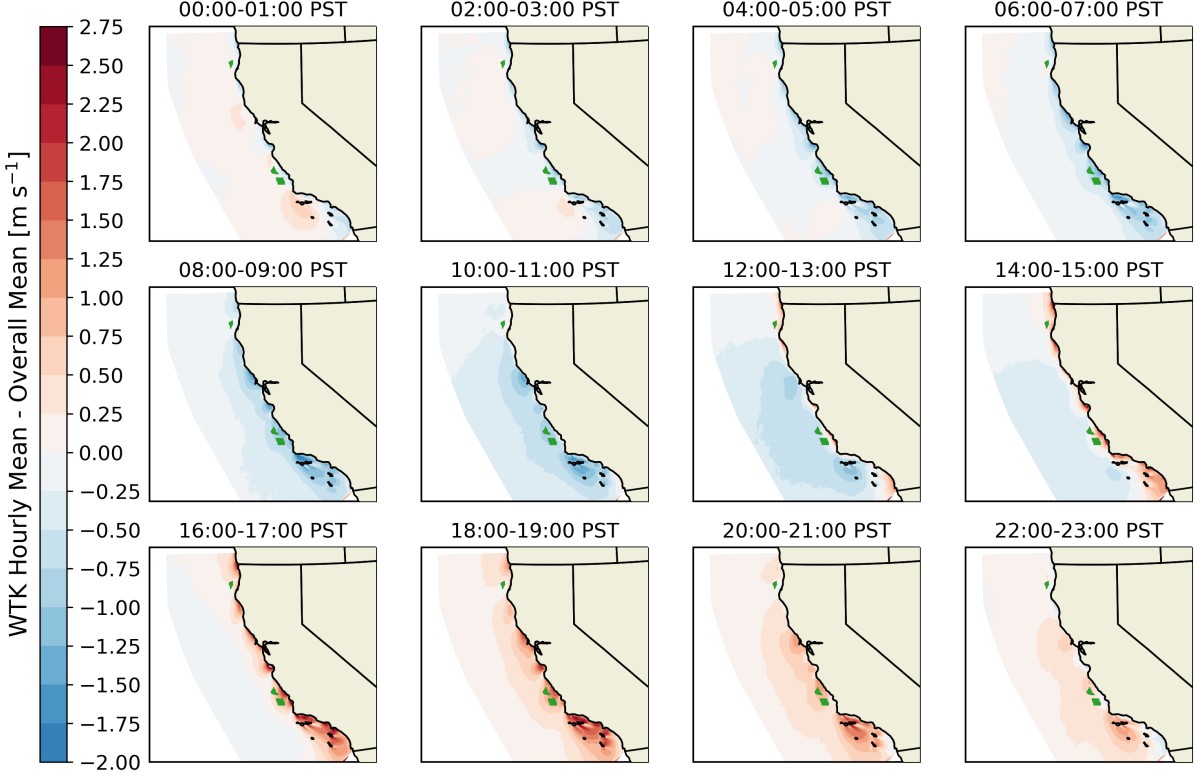

**Figure B4.** Hourly averaged 100-m WIND Toolkit winds, with reference to the overall WIND Toolkit mean.



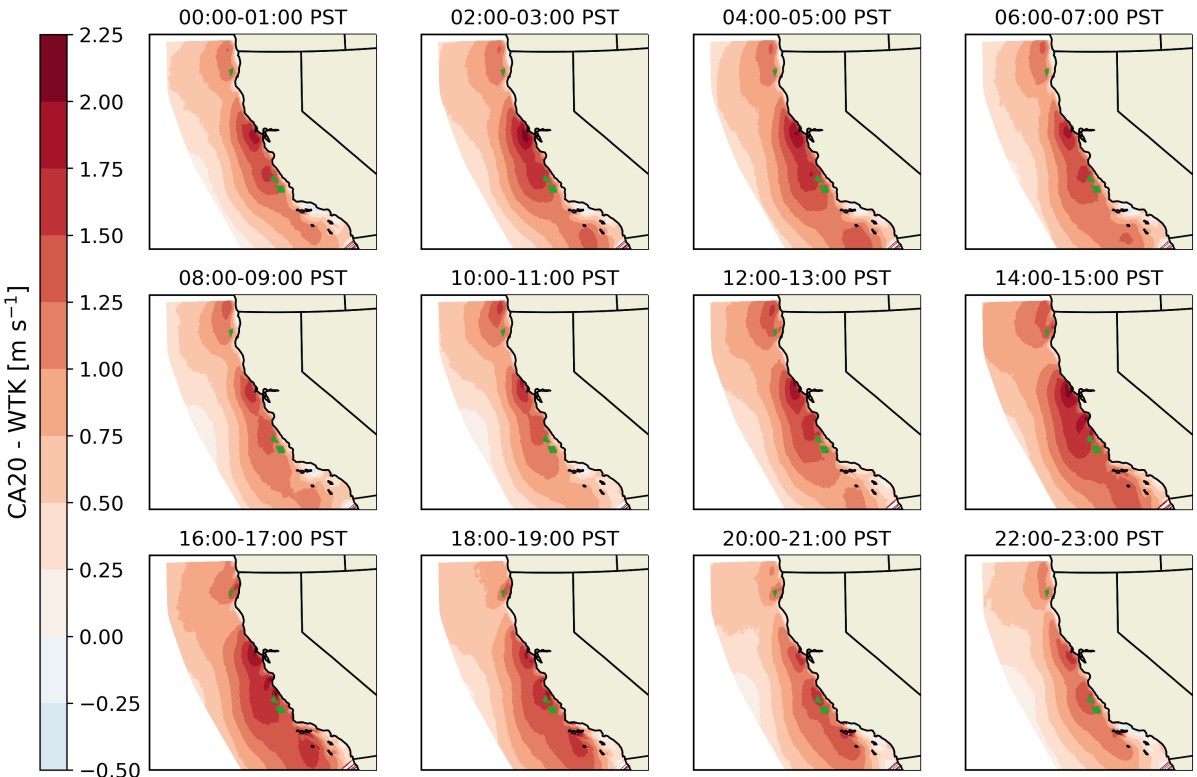

**Figure B5.** Difference in hourly averaged 100-m winds between CA20 and the WIND Toolkit. Every other hour is shown for concision.



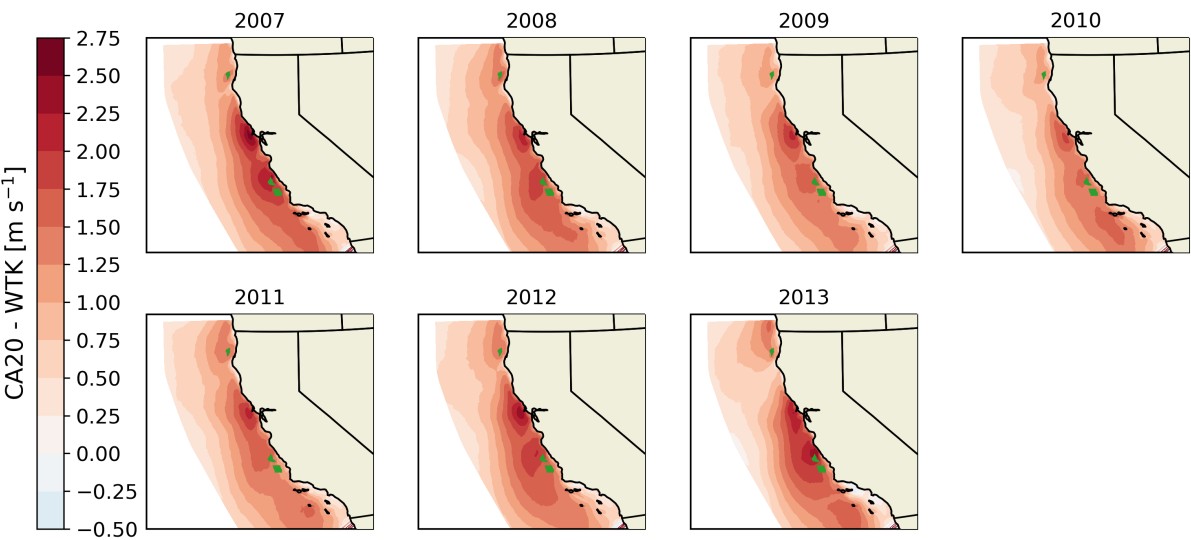

**Figure B6.** Difference in annually averaged 100-m winds between CA20 and the WIND Toolkit.



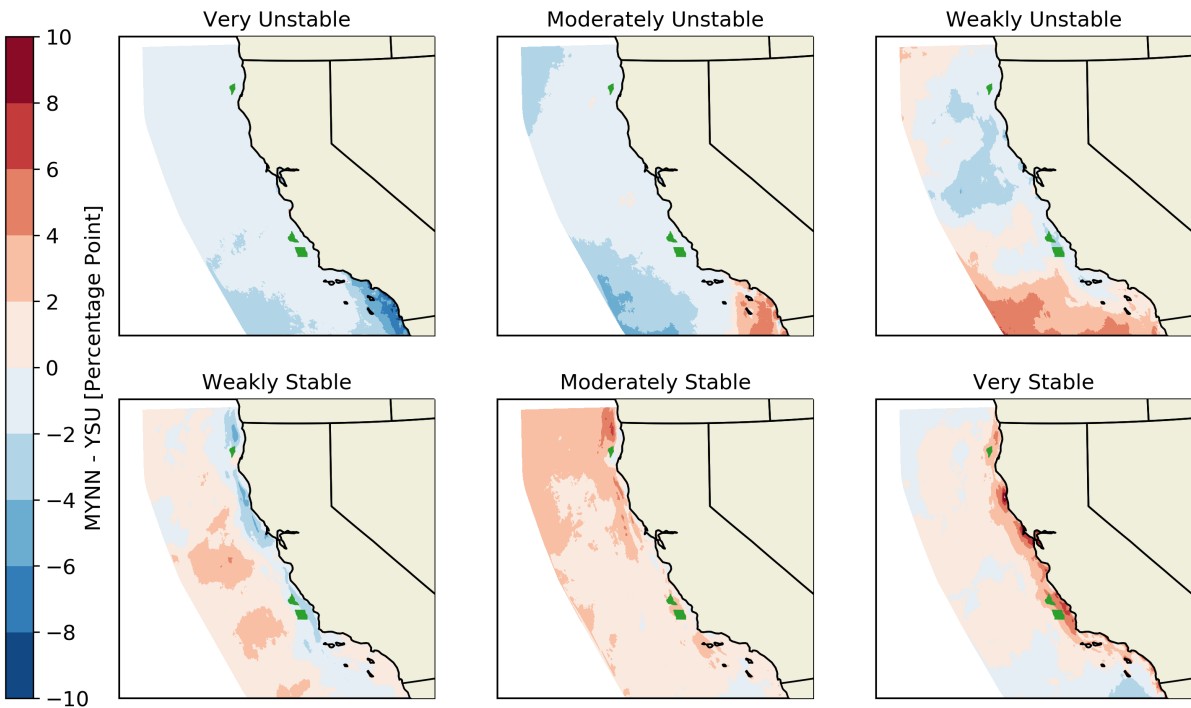

**Figure B7.** Difference in percentage frequency in each stability between MYNN and YSU.





*Author contributions.* MO and WM conceptualized the study. AR and MO conducted the simulations. MR post-processed raw WRF data. AR, JKL, and MO conducted analysis and contributed to writing.

*Competing interests.* The authors declare that they have no competing interests.

*Acknowledgements.* This work was supported and funded by the Bureau of Ocean Energy Management (BOEM) under Agreement No. IAG-19-2123.





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
