# Peer review of "A Twenty-Year Analysis of Winds in California for Offshore Wind Energy Production Using WRF v4.1.2"

_Geoscientific Model Development, 2021_

## Author Comment (AC1)

**A Twenty-Year Analysis of Winds in California for Offshore Wind Energy Production Using WRF v4.1.2**

**Review 1**

Dear Reviewer, thank you for taking the time to review our manuscript and thank you for the insightful feedback. We have provided an itemized response below, where our comments are marked in red.

**General comments:**

This paper deals with the differences in the modelled datasets between two different model setups. I would expect two different setups to produce different results, however, it is currently unclear which one of these setups is better, as the comparison with observations is not yet available. Regrettably, the performance of the PBL scheme near the surface (e.g. when compared to buoys) is not indicative of the performance at the hub heights. Moreover, even if we look at the verification results for buoys (Table A1) it is hard to argue that one setup is better than the other. The paper argues that the differences in wind-speed results from different PBL schemes can be explained by differences in frequency between different atmospheric stability classes. Do we know which of the PBL schemes provides a better (closer to observations) description of stability? Not at this point, regrettably. In summary, I am afraid that the lack of comparison with hub height observations diminishes the applicability of the conclusions carried out in this paper.

In this manuscript, we primarily conduct a model inter-comparison study. We believe these types of studies to be essential, especially for trying to understand uncertainty of modeled wind resource (Research Need #2, from Archer et. al 2014). Additionally, model inter-comparison studies are explicitly called for in the "model experiment description papers" category in GMD (fifth bullet at https://www.geoscientific-model-development.net/about/aims\_and\_scope.html).

However, we additionally agree that it valuable and timely to compare to the simulations to hub-height wind speed measurements from lidar, which have only recently become available. In this spirit, we conducted a limited validation study for the month of October 2020 that maintains the model-focused nature of this analysis while also increasing the applicability of the conclusions of this paper. This analysis is detailed in the new Appendix B (L 427-447), and we summarize the analysis here.

We use measurements from two lidars---one deployed off the coast of Humboldt and the other near Morro Bay. As neither CA20 nor the WIND Toolkit contain wind information for October 2020, we run two new month-long simulations. The first simulation uses the same set up as CA20, and the second simulation uses the same set up except YSU is used instead of MYNN, to correspond to the WIND Toolkit setup.

Figure R1: Profiles of wind speed bias and RMSE at Humboldt and Morro Bay for the October 2020 MYNN and YSU simulations.

We find that, for this month, MYNN outperforms YSU (Fig. R1). In Humboldt, MYNN wind speed profiles show a bias between 1.0 and 2.5 m s-1. The YSU wind speed profiles have a bias that is approximately 0.5 m s-1 larger at all heights. MYNN also shows a smaller RMSE than YSU here. Additionally, MYNN outperforms YSU at Morro Bay in terms of bias and RMSE, although by smaller quantities. The difference in performance at the two locations may be tied to modeled stability: the models at Humboldt show predominantly weakly stable and moderately stable values of the bulk Richardson number, whereas they show a greater spread of stabilities at Morro Bay (Fig. R2).

Figure R2: Bulk Richardson numbers at the location of the Humboldt and Morro Bay lidars for the MYNN and YSU simulations in October 2020.

Thus, this initial comparison suggests that MYNN is more accurate than YSU at these two locations, and therefore, it could be postulated that CA20 is more accurate than the WIND Toolkit. Future studies will expand this analysis beyond a month to study modeled winds under a longer observational window.

Specific comments (major)

- Changes in the MYNN PBL scheme: "The WIND Toolkit was developed using a 7-year (2007–2013) simulation with WRF 3.4.1. CA20 builds upon this by using WRF 4.1.2 across a 20-year period (2000–2019)." (Line 76-77). CA20 uses the MYNN parametrization scheme, the WIND Toolkit uses the YSU scheme (Lines 85-89). The problem is that in WRF version 3.7 the MYNN scheme underwent significant changes, and indeed the authors acknowledge this (Line 192). The thing that I do not understand is why if the WRF version is one of the factors that is analyzed during the sensitivity study, why aren't the changes in the MYNN parametrization scheme also included in the sensitivity analysis (as a separate parameter)? There are no methodological difficulties, one would just need to run both WRF versions with the MYNN scheme, instead of the YSU scheme, as it was already done. The reason why I would like to see such analysis is that changes in the MYNN scheme can lead to significantly worse verification results at hub heights when compared to observations (see Figure 8, section 5.3 in Hahmann et al. 2020).
- Our sensitivity study (summarized in Figure 6 of the manuscript) aims to study the factors that have changed between the WIND Toolkit and CA20. As noted, CA20 uses MYNN and WRF 4.2.1 whereas the WIND Toolkit uses YSU and WRF 3.4.1. While MYNN saw major changes in 3.7.1, the winds in the WIND Toolkit come from YSU and not MYNN, so we do not believe this update to the MYNN PBL scheme is directly relevant for this sensitivity study.

Nonetheless, other studies (such as Hahmann et al. 2020) have shown that the change to MYNN in WRF 3.7.1 can significantly impact hub-height wind. To contribute to this scientific discourse, we additionally simulated a month of winds in October 2020 using WRF 3.4.1 and MYNN. This additional month of simulation enables comparison to the aforementioned lidar measurements.

Figure R3: Same as Figure R1, except with a MYNN WRF v3.4.1 run.

---

## Author Comment (AC2)

**A Twenty-Year Analysis of Winds in California for Offshore Wind Energy Production Using WRF v4.1.2**

**Review 2**

Dear reviewer, thank you for volunteering your time to review our manuscript and to provide comments, and thank you for the kind words as well. Below, we incorporate and respond to the feedback that you have provided.

The manuscript describes a new model-generated dataset of wind resources for the California coast and compares it to an older dataset used by the wind industry. The manuscript is well written and well organized. The manuscript presents many interesting statistics of the wind climatology in these two datasets, e.g., the climatology of wind shear, wind veer, and "wind droughts". There are no observational datasets of wind in this region above the surface (buoys); thus, the comparison is purely made between two model-generated datasets. One could probably argue that the most recent one is more accurate, but the manuscript shows no evidence that this is true. The appendix contains a short evaluation against buoy data. But the authors well know that this is insufficient because the different surface and PBL scheme could give very different wind profiles (see, e.g., Draxl et al. 2014). My usual question for this type of manuscript is: "what new information does the manuscript provide that will help the scientific community in future investigations?" I cannot find any. Two datasets are compared, they are different in many aspects, but they cannot guide future WRF simulations for wind resource assessment. The information is perhaps valuable for wind farm developers and policymakers, but in my opinion, not to the reader of GMD.

Therefore, my recommendation is that the manuscript is rejected for publication in GMD but perhaps transferred to Wind Energy Science.

In this manuscript, we primarily conduct a model inter-comparison study. We believe these types of studies to be essential, especially for trying to understand uncertainty of modeled wind resource (Research Need #2, from Archer et. al 2014). Model inter-comparison studies are explicitly called for in the "model experiment description papers" category in GMD (fifth bullet at https://www.geoscientific-model-development.net/about/aims_and_scope.html), and as such, we believe this manuscript is an appropriate fit for this journal.

However, we additionally agree that it valuable and timely to compare to the simulations to hub-height wind speed measurements from lidar, which have only recently become available. In this spirit, we conducted a limited validation study for the month of October 2020 that maintains the model-focused nature of this analysis while also increasing the applicability of the conclusions of this paper. This analysis is detailed in the new Appendix B (L 427-447), and we summarize the analysis here.

We use measurements from two lidars---one deployed off the coast of Humboldt and the other near Morro Bay. As neither CA20 nor the WIND Toolkit contain wind information for October 2020, we run two new month-long simulations. The first simulation uses the same set up as CA20, and the second simulation uses the same set up except YSU is used instead of MYNN, to correspond to the WIND Toolkit setup.

[Figure]

Figure R1: Profiles of wind speed bias and RMSE at Humboldt and Morro Bay for the October 2020 MYNN and YSU simulations.

We find that, for this month, MYNN outperforms YSU (Fig. R1). In Humboldt, MYNN wind speed profiles show a bias between 1.0 and 2.5 m s$^{-1}$. The YSU wind speed profiles have a bias that is approximately 0.5 m s$^{-1}$ larger at all heights. MYNN also shows a smaller RMSE than YSU here. Additionally, MYNN outperforms YSU at Morro Bay in terms of bias and RMSE, although by smaller quantities. The difference in performance at the two locations may be tied to modeled stability: the models at Humboldt show predominantly weakly stable and moderately stable values of the bulk Richardson number, whereas they show a greater spread of stabilities at Morro Bay (Fig. R2).

[Figure]

Figure R2: Bulk Richardson numbers at the location of the Humboldt and Morro Bay lidars for the MYNN and YSU simulations in October 2020.
Thus, this initial comparison suggests that MYNN is more accurate than YSU at these two locations, and therefore, it could be postulated that CA20 is more accurate than the WIND Toolkit. Future studies will expand this analysis beyond a month to study modeled winds under a longer observational window.

Furthermore, we have received your feedback that you cannot find any information that answers the question "what new information does the manuscript provide that will help the scientific community in future investigations?".

- One major question that the scientific community is investigating is "How sensitive are modeled hub-height winds to different inputs in numerical weather prediction models?" This question has been addressed in GMD articles (see Hahmann et al. (2020)), as well as in other journals that are less application-driven and distant from wind industry (see Hahmann et al. (2015), Yang et al. (2017), and Berg et al. (2019)). Section 3 of our paper directly responds to this question by analyzing average wind speeds across the coast. Sections 3.1--3.3 address *how* the winds are different, and

Section 3.4 investigates *why* the winds may be different. Sections 4.1 (Wind Speed Distributions), 4.3 (Stability Analysis), and 4.4 (Wind Shear and Veer) additionally directly address this question. Ultimately, our results in offshore California suggest that average modeled hub-height winds are not particularly sensitive to the period of simulation, forcing from the reanalysis or SST product, and the horizontal domain extent (Figure 6 c-f). In contrast, we find that hub-height winds are sensitive to the PBL scheme and to the WRF version. As discussed in greater detail within the manuscript (L 169-170, L 199-201, L 218-219, L223-224, L227), our findings agree with some scientific studies but disagree with others. As a result, we believe that our study will help guide future scientific investigations, particularly into these areas of disagreement.

- Furthermore, we disagree with the statement that this study "cannot guide future WRF simulations for wind resource assessment" as our study has already informed future scientific investigations. Since this pilot study has been conducted, the National Renewable Energy Laboratory has conducted additional offshore wind resource assessments across many regions of the US including the Pacific Northwest and the Mid-Atlantic (manuscripts in preparation). The modeling decisions made in these new assessments were directly informed by the results of this study.

Also, I have a few more editorial comments:

1. Please revise the figure captions. Most figure captions need further clarification. Many of them lack information on the averaging period. Also, please add (a), (b) labels to all sub-panels. These labels are a requirement from GMD.

We have revisited figure captions and have added averaging period information where it was missing (Fig. 6, Fig. 9). Additionally, all sub-panels have been labeled.

2. Abstract L5-6: "The data set predicts a significantly larger wind resource (0.25–1.75 m s−1 stronger)," Since the units are m s-1. This is wind speed, not wind resource.

Thank you for catching this inconsistency. We have updated the abstract to read "The data set predicts significantly larger mean hub-height wind speeds (0.25–1.75 m s−1)" (L 6).

3. Page 4, the bottom of page: "CA20 applies spectral nudging on a 6-km domain every 6 hours" This statement is incorrect. In WRF, the nudging terms are applied at every time step. But the tendencies used to compute this term could come to 6 hourly data.

Thank you for catching this. The corrected statement now reads "CA20 applies spectral nudging on a 6-km domain using tendencies that are updated at 6 hourly resolution." (L 94).

4. Page 10, L180. Is the MYNN simulation used as the basis for the stability classes? I think this needs to be clarified.

We have clarified the situation, as stability classes are calculated for both MYNN and YSU. The introduction to this analysis now reads "We diagnose stability *in each of the PBL schemes* via the bulk Richardson number in the lowest 200 m" (L 173, italics is new text).

5. L210: You write, "The updated product contains higher horizontal resolution (31 km vs 79 km), higher temporal resolution (1 hour vs 6 hours)". But I understand that you use the 6-hour ERA5 data for the nudging, so this fact is inconsequential.

Thank you for drawing attention to this detail. We generated ERA5 boundary conditions (*met_em* files) using hourly data. However, we applied spectral nudging a 6 hourly resolution as we did not want to nudge too frequently. We have updated this text to now read "The updated product contains higher horizontal resolution (31 km vs. 79 km), higher temporal resolution (1 hour vs. 6 hour for boundary conditions)" (L 215-216).

6. I don't see the point of including section 3.4.6. The different factors are clearly interrelated and nonlinear. So why analyze their sum?

Section 3.4.6 provides a linear understanding of sensitivity to all the inputs. Based off of the lead author's coursework in a graduate-level Uncertainty Quantification course at CU Boulder (taught by Dr. Paul Constantine based off of the professor's personal notes), this is the typical first step before conducting a larger, non-linear sensitivity study.

References:

Archer, C. L., and Coauthors, 2014: Meteorology for Coastal/Offshore Wind Energy in the United States: Recommendations and Research Needs for the Next 10 Years. *Bull. Amer. Meteor. Soc.*, **95**, 515–519, https://doi.org/10.1175/BAMS-D-13-00108.1.

Berg, L. K., Y. Liu, B. Yang, Y. Qian, J. Olson, M. Pekour, P.-L. Ma, and Z. Hou, 2019: Sensitivity of Turbine-Height Wind Speeds to Parameters in the Planetary Boundary-Layer Parametrization Used in the Weather Research and Forecasting Model: Extension to Wintertime Conditions. *Boundary-Layer Meteorology*, **170**, 507–518, https://doi.org/10.1007/s10546-018-0406-y.

Hahmann, A. N., C. L. Vincent, A. Peña, J. Lange, and C. B. Hasager, 2015: Wind climate estimation using WRF model output: method and model sensitivities over the sea. *International Journal of Climatology*, **35**, 3422–3439, https://doi.org/10.1002/joc.4217.

Hahmann, A. N., and Coauthors, 2020: The Making of the New European Wind Atlas, Part 1: Model Sensitivity. *Geoscientific Model Development Discussions*, 1–33, https://doi.org/10.5194/gmd-2019-349.

Yang, B., and Coauthors, 2017: Sensitivity of Turbine-Height Wind Speeds to Parameters in Planetary Boundary-Layer and Surface-Layer Schemes in the Weather Research and Forecasting Model. *Boundary-Layer Meteorology*, **162**, 117–142, https://doi.org/10.1007/s10546-016-0185-2.